# Modelling Complex Tabular Datasets with a Mixture of Diverse Generative Models

**Antoine Faul**[*]                                                   *antoine.faul@unibe.ch*
*Institute of Mathematical Statistics and Actuarial Sciences*
*University of Bern*

**Xiao Zhou**[*]                                                   *xiao.zhou.23@alumni.ucl.ac.uk*
*Department of Computer Science*
*University College London*

**Ossi Räisä**[†]                                                   *ossi.raisa@cispa.de*
*CISPA Helmholtz Center for Information Security*

**Mihaela Van der Schaar**                                                   *mv472@cam.ac.uk*
*University of Cambridge*

**Cem Tekin**                                                   *cemtekin@ee.bilkent.edu.tr*
*Bilkent University*

**Reviewed on OpenReview:** *https://openreview.net/forum?id=3y3mHAldp7*

## Abstract

Generative models are widely used, yet they often struggle to capture the multi-modal structure of complex tabular datasets. We address this challenge by introducing a novel framework that employs mixtures of diverse generators, each specialized to different regions of the data space. Our method proceeds in two stages: first, generators are assigned to data clusters via a compute-efficient bandit-based allocation strategy; second, cluster assignments are refined through an iterative procedure inspired by the Expectation–Maximization (EM) framework. Crucially, our approach is designed for settings where the generators' likelihoods are intractable and only generated data samples are accessible. In a simpler setting where clusters can be approximately identified, we derive theoretical results based on the robustness of the Maximum Mean Discrepancy (MMD). Empirical evaluations on both synthetic and real-world tabular datasets demonstrate that our approach produces high-quality synthetic data, validating its effectiveness in challenging generative modeling tasks.

## 1 Introduction

Generative models have revolutionized machine learning by enabling the creation of realistic, data-like outputs from input datasets. These models excel at capturing the underlying patterns of training data, allowing them to generate new, synthetic samples for a wide range of applications—a process known as synthetic data generation. Recent breakthroughs have led to the development of advanced deep generative models like Diffusion models (Azizi et al., 2023), Generative Adversarial Networks (GANs) (Goodfellow et al., 2020), and Variational Autoencoders (VAEs) (Kingma & Welling, 2013). These models excel with complex structured data types, such as text and images, and have drawn significant interest from the machine learning community.

---

*Equal contribution. [†]Work done primarily at University of Helsinki.
Code available at: https://github.com/SkynetMonster/BIRD

While tabular data is critical across various domains, advancements in generating synthetic tabular data have lagged (Manousakas & Aydöre, 2023). However, this area is now gaining more attention. Adaptations of the aforementioned models, such as CTGAN and TVAE (Xu et al., 2019), have shown promise in producing high-quality synthetic data. Additionally, a variety of other synthetic data generators, based on Diffusion Models (Kim et al., 2023; Shi et al., 2025), Large Language Models (LLMs) (Zhao et al., 2025), and Random Forests (Watson et al., 2023; Nock & Guillame-Bert, 2024), have emerged in the literature.

However, the intrinsic multi-modal and complex nature of real-world data poses challenges such as mode collapse (Lala et al., 2018), where models excel in certain distribution areas but fail to capture others comprehensively. To circumvent this issue, we explore the promising approach of combining diverse generative models to leverage the strengths of each generator while ensuring diversity in the synthetic data produced. By allowing each generator to specialize in a specific data region, we aim to capture the multi-modalities in the dataset and to produce high-quality samples.

While optimizing mixture weights for fixed generators using metrics like Maximum Mean Discrepancy (MMD) is relatively straightforward, the real challenge lies in simultaneously learning these data regions, denoted as clusters, and their corresponding generators from a dataset. To address this challenge, we first offer theoretical insights into situations where data clusters are imperfectly learned. In a simplified scenario where correct clusters can be roughly identified, we demonstrate that an optimal mixture of generators can be learned up to a certain accuracy by leveraging robustness properties of the MMD. However, in more realistic scenarios where clusters are unknown and difficult to approximate, we introduce a two-step algorithm to tackle this issue and efficiently combine diverse generators.

The first step employs a bandit-inspired method to assign generators to data clusters in the best possible way given compute constraints, and the second step involves an iterative cluster adjustment mechanism inspired by the EM framework to fine-tune the performance. This approach is tailored for scenarios where generators have intractable likelihoods, yet allow for easy sampling of points.

An advantage of our method is its ability to accurately represent underrepresented classes in real-world datasets. We demonstrate this by synthesizing a subset Credit Card Fraud Detection dataset containing unbalanced classes with only 2% in the minority class. Table 1 shows that our algorithm, BIRD (see Section 4 for its description), outperforms complex individual generators by accurately sampling the correct proportion of minority class points and identifying distributions in both classes.

|  | BIRD | ARF | CTGAN | TabDDPM | NFlow | TVAE |
|---|---|---|---|---|---|---|
| **Majority Prop.* (%)** | **97.9** | 98.4 | 98.6 | 94.3 | 96.9 | 99.9 |
| **P-F1 Majority** ↑ | **0.640** | 0.621 | 0.199 | 0.482 | 0.367 | 0.427 |
| **P-F1 Minority** ↑ | **0.686** | 0.548 | 0.141 | 0.372 | 0.187 | † |

Table 1: Performance of synthetic data generators evaluated by P-F1 scores for majority and minority classes.[1]*True proportion is 98%. † indicates insufficient data for metric estimation. Bold values highlight the best performance.

Our contributions are as follows:

1. We demonstrate that under mild assumptions, leveraging the robustness properties of the MMD, learning a mixture of distributions from imperfect clusters yields an error rate of $\mathcal{O}(n^{-1/2})$ with $n$ as the sample size. This rate is comparable to the scenario where clusters are perfectly identified.

2. We propose a bandit-based method that efficiently learns the best generator for each data cluster under a given compute budget.

3. We introduce an EM-inspired algorithm to train mixtures of diverse, intractable generative models by iteratively refining the initial clusters and updating the generators' parameters.

---

[1]Additional details about the experimental setup and metrics can be found in Appendix H.

    4. We illustrate the practical efficacy of our approach through empirical results on both simulated and real-world datasets.

## 2 Related work

Numerous studies have explored the use of mixtures of generative models to effectively capture multi-modal datasets. Many focus on modifying the training procedures or architectures of well-established generative models like GANs to address mode collapse ((Eghbal-zadeh et al., 2019; Ghosh et al., 2018)). For instance, Park et al. (2018) trains a mixture of GANs using a gating network to learn cluster assignments. In the context of VAEs, Shi et al. (2019) employs a mixture of experts to create a multi-modal variational posterior.

Some other approaches leverage pretrained generators. Rezaei et al. (2025) propose an online algorithm to optimize mixture weights, whereas our method emphasizes *simultaneous learning* of clusters and generators, resulting in improved clustering accuracy. Banijamali et al. (2017) also investigate simultaneous learning of clusters and generators through an EM-like algorithm. Unlike our work, their approach is limited to mixtures of the same type of generators and relies on random initial cluster-generator assignments. We introduce a bandit-based approach to optimize initial cluster-generator assignments, effectively leveraging *diverse generators* within our mixture.

Training mixtures of generative models with boosting algorithms has been suggested by Tolstikhin et al. (2017), but this sequential training is time-intensive. To mitigate this, Locatello et al. (2018) offer a competitive training procedure with an additional discriminator, allowing parallel training. However, our approach uniquely remains *budget-aware*, allowing users to effectively control the computational budget at each step of the algorithm, which is not possible with the other methods.

Table 2 compares our method with closely related works, and highlights several key properties discussed earlier.

| Method | Simultaneous learning | Diverse generators | Budget aware |
|---|:---:|:---:|:---:|
| Banijamali et al. (2017) | ✓ | ✗ | ✗ |
| Rezaei et al. (2025) | ✗ | ✓ | ✗ |
| Locatello et al. (2018) | ✓ | ✗ | ✗ |
| BIRD (Ours) | ✓ | ✓ | ✓ |

Table 2: Comparison of the properties of different methods for training mixtures of generative models.

## 3 Learning mixtures of diverse generative models

We consider having access to i.i.d. observations $(x_1, \ldots, x_n) \in \mathcal{X}^n$ drawn from a random variable $X$ taking values in a space $\mathcal{X}$. We assume $X$ follows a finite mixture distribution $\mathbb{P}_{\mathrm{mix}} = \sum_{i=1}^{J} \lambda_i \mathbb{P}_i$, where $J \geq 1$, $\lambda_i \geq 0$ for $i \in [J]$, and $\sum_{i=1}^{J} \lambda_i = 1$. In each component, we aim to approximate $\mathbb{P}_i$ by a parametric distribution $\mathbb{P}_{\theta_i}$, referred to as a generator throughout this paper, where $\theta_i$ belongs to a parameter set $\Theta_i$ for $i \in [J]$. Since each individual distribution $\mathbb{P}_i$ can exhibit significantly different characteristics, we require a diverse set of probability distributions to effectively capture their unique properties. Our goal is to learn this mixture of diverse generative models.

For our theoretical results, we focus on the Maximum Mean Discrepancy (MMD) (Gretton et al., 2012) as the distance between two probability distributions to minimize. Given a characteristic kernel $k$, the squared MMD between two distributions $\mathbb{P}$ and $\mathbb{Q}$ is defined by

$$\mathrm{MMD}_k^2(\mathbb{P}, \mathbb{Q}) = \mathbb{E}_{X,X'\sim\mathbb{P}}[k(X,X')] + \mathbb{E}_{Y,Y'\sim\mathbb{Q}}[k(Y,Y')] - 2\mathbb{E}_{X\sim\mathbb{P},Y\sim\mathbb{Q}}[k(X,Y)] \, .$$

One motivating factor for using the MMD as an evaluation metric for mixture of generative models is its convexity with respect to the first argument (Allen et al., 2025). This convexity result suggests that a mixture of generative models inherits the average performance of its individual components. Specifically, the mixture will perform at least as well as the poorest-performing generator within it, regardless of the mixing proportions. If we can identify the optimal weights that minimize the MMD to the target distribution, the mixture will perform at least as well as the best standalone generator in the mixture.

**Remark 3.1** *As demonstrated in works such as Rezaei et al. (2025) and Allen et al. (2025), optimizing the weights with respect to the empirical distribution of $(x_1, \ldots, x_n) \in \mathcal{X}^n$ is a quadratic convex problem that can be solved exactly with specific solvers (interior point methods, active-set methods, etc.)*

### 3.1 Learning generators on imperfect clusters

The aim of our study is to optimally learn a mixture of distributions when the generators are not fixed, necessitating simultaneous estimation of both generators' parameters and mixing proportions.

When observations are sampled from a mixture, having access to the latent variable indicating which component each point originates from would naturally lead to training each generator on distinct clusters. The generators would then be combined, assigning weights based on the proportion of points in each cluster, allowing each generator to specialize in specific regions of the input space. This strategy has been explored in the literature, offering privacy guarantees for the resulting mixture (Acs et al., 2018).

Yet, in practical situations, the latent variables remain unknown, preventing precise recovery of the underlying clusters. Under mild assumptions, and leveraging the robustness properties of the MMD, we show that learning a mixture of distributions from imperfect clusters achieves an error rate comparable to that of the well-specified setting, in which the clusters are perfectly identified. We emphasize, however, that these results are derived in a simpler setting than the algorithm we present later in Section 4. We introduce the relevant notation in the following paragraph.

**Clustering and Empirical Measures:** Given $J \in \mathbb{N}^*$ and our i.i.d. samples $x = (x_1, \ldots, x_n) \in \mathcal{X}^n$, a clustering rule $c$ partitions these observations into $J$ sets. Formally, the clustering rule $c$ is represented as $c = (x^{(1)}, \ldots, x^{(J)})$, where $x^{(i)} \in \mathcal{X}^{n_i}$ for all $i \in [J]$, satisfying $\sum_{i=1}^{J} n_i = n$ and $n_i > 0$. Additionally, we ensure that

$$\bigcup_{i=1}^{J} x^{(i)} = x \ \text{ and } \ x^{(i)} \cap x^{(j)} = \emptyset \ \forall i \neq j \ .$$

The partition results in $J$ *clusters* denoted as $x^{(i)}$ for $i \in [J]$.

We denote $\hat{P}$ the empirical distribution of our i.i.d. samples $x$ from $\mathbb{P}_{\text{mix}}$. We assume that we know from which distribution each point has been generated, in other words, the true clustering rule $c^* = (x^{*(1)}, \ldots, x^{*(J)})$ is known.

We apply a clustering algorithm to recover these true clusters, yielding an estimated clustering rule $\tilde{c} = (\tilde{x}^{(1)}, \ldots, \tilde{x}^{(J)})$. We assume that the estimated clusters $\tilde{x}^{(i)}$ match the true cluster sizes $n_i$, with their empirical distribution given by

$$\tilde{P}_i = \frac{1}{n_i} \sum_{\tilde{x}_l \in \tilde{x}^{(i)}} \delta_{\tilde{x}_l} \ ,$$

where $\delta_{\tilde{x}_l}$ denotes the Dirac measure concentrated at the point $\tilde{x}_l$.

The misclassification rate for the $i$th cluster is defined as the cardinality of the symmetric difference $x^{*(i)}$ and $\tilde{x}^{(i)}$, divided by $2n_i$, i.e.,

$$\text{msr}(x^{*(i)}, \tilde{x}^{(i)}) = \frac{|x^{*(i)} \triangle \tilde{x}^{(i)}|}{2n_i} \ .$$

We define the misclassification rate of the clustering rule $\tilde{c}$ with respect to the true clustering rule $c^*$ as

$$\text{msr}(c^*, \tilde{c}) = \max_{i \in \{1, \ldots, J\}} \text{msr}(x^{*(i)}, \tilde{x}^{(i)}) \ .$$

We consider the following assumptions:

**Assumption 3.2** *The problem is well-specified within each true cluster, i.e., for all $i \in [J]$, there exists $\theta_{0,i} \in \Theta_i$ such that $\mathbb{P}_i = \mathbb{P}_{\theta_{0,i}}$, where $\Theta_i \subseteq \mathbb{R}^{a_i}$ with $a_i$ the number of parameters of the distribution $\mathbb{P}_i$.*

**Assumption 3.3** *The misclassification rate of the clustering rule $\tilde{c}$ with respect to the true clustering rule $c^*$ is smaller than $\epsilon_n > 0$.*

**Assumption 3.4** *In each identified cluster $\tilde{x}^{(i)}, i \in [J]$ of size $n_i$, we are able to find a parameter $\tilde{\theta}_{n_i}^{(i)}$, approximately minimizing the MMD up to a slackness of $\chi_n \geq 0$, with respect to points in cluster $i$, such that*

$$\mathrm{MMD}_k\left(\mathbb{P}_{\tilde{\theta}_{n_i}^{(i)}}, \tilde{P}_i\right) \leq \inf_{\theta \in \Theta_i} \mathrm{MMD}_k\left(\mathbb{P}_\theta, \tilde{P}_i\right) + \chi_n .$$

Assumption 3.2 asserts that our generator family is rich enough to include the correct probability distributions for each cluster. Assumption 3.3 posits that the true clusters can be accurately identified, subject to a fixed misclassification rate. Finally, Assumption 3.4 states that within each cluster, it is feasible to approximately find generator's parameters that minimize the MMD with respect to the empirical distributions.

Assuming we can precisely estimate the optimal weights $\hat{\lambda} = \{\hat{\lambda}_i\}_{i=1}^J$ that minimize the MMD relative to the empirical distribution as explained in Remark 3.1, i.e.,

$$\hat{\lambda} \in \underset{\lambda=(\lambda_1,\ldots,\lambda_J)}{\operatorname{argmin}} \ \mathrm{MMD}\left(\hat{P}, \sum_{i=1}^J \lambda_i \mathbb{P}_i\right) \quad \text{s.t.} \quad \lambda_i \geq 0, \ \sum_{i=1}^J \lambda_i = 1 ,$$

we can apply the generalization bound from Theorem 3.1 in Chérief-Abdellatif & Alquier (2022), which guarantees that

$$\mathbb{E}(\mathrm{MMD}_k(\mathbb{P}_{\hat{\lambda}}, \mathbb{P}_{\mathrm{mix}})) \leq 2\sqrt{\frac{1}{n}}, \quad \text{with } \mathbb{P}_{\hat{\lambda}} = \sum_{i=1}^J \hat{\lambda}_i \mathbb{P}_i .$$

where the expectation is taken over the sampling of $x$ from $\mathbb{P}_{\mathrm{mix}}$. Furthermore, by utilizing the robustness of MMD against outliers as demonstrated in Proposition 3.5 of Chérief-Abdellatif & Alquier (2022), we establish the following finite-sample consistency guarantees:

**Corollary 3.5** *If Assumptions 3.2, 3.3 and 3.4 hold, with i.i.d. samples $x = (x_1, \ldots, x_n)$ from $\mathbb{P}_{mix}$, the kernel $k$ is characteristic and bounded by 1, then there exists weights $\tilde{\lambda} = \{\tilde{\lambda}_i\}_{i=1}^J$ such that*

$$\mathbb{E}\left(\mathrm{MMD}_k\left(\sum_{i=1}^J \tilde{\lambda}_i \mathbb{P}_{\tilde{\theta}_{n_i}^{(i)}}, \mathbb{P}_{mix}\right)\right) \leq 4\epsilon_n + \sum_{i=1}^J 4\lambda_i \mathbb{E}\left(\frac{1}{\sqrt{n_i}}\right) + \chi_n + \frac{2}{\sqrt{n}} ,$$

*where the expectation is computed over samples drawn from the empirical distributions. Furthermore, with $0 < \gamma < 1$, we have*

$$\mathbb{P}\left(\mathrm{MMD}_k\left(\sum_{i=1}^J \tilde{\lambda}_i \mathbb{P}_{\tilde{\theta}_{n_i}^{(i)}}, \mathbb{P}_{mix}\right) \leq \frac{4}{\gamma}\left(\epsilon_n + \frac{\chi_n}{4} + \frac{1}{2\sqrt{n}} + \sum_{i=1}^J \mathbb{E}\left(\frac{\lambda_i}{\sqrt{n_i}}\right)\right)\right) \geq 1 - \gamma .$$

The proof of Corollary 3.5 is given in Appendix A.

With $J$ fixed as $n$ becomes large, if the size of each cluster verifies $n_i \geq cn$ with a constant $c > 0$ for $i \in [J]$, the misclassification rate satisfies $\epsilon_n = \mathcal{O}(n^{-\frac{1}{2}})$ and the precision in learning the true parameter within each cluster meets $\chi_n = \mathcal{O}(n^{-\frac{1}{2}})$, we establish a finite sample error bound of order $n^{-1/2}$, akin to the well-specified case (Chérief-Abdellatif & Alquier, 2022), even when the true clusters in the data are not precisely identified. However, in practice, identifying even approximate clusters can be challenging.

In the case of unbalanced cluster sizes, the result still holds. However, if for some $i_0 \in \{1, \ldots, J\}$, $n_{i_0} = \mathcal{O}(1)$ as $n \to +\infty$, then the rate of $\mathcal{O}(n^{-\frac{1}{2}})$ is no longer satisfied.

In the next section, we propose an algorithm that addresses the issue of identifying clusters precisely by iteratively updating the clusters and generators simultaneously, after an initial clustering step.

## 4 BIRD algorithm

Accurately estimating parameters within a mixture of distributions is a well-studied yet challenging task due to issues such as non-identifiability (Teicher, 1963). While the Expectation-Maximization (EM) algorithm (Dempster et al., 1977) remains a popular method for iteratively maximizing likelihood in the presence of latent variables, optimizing such models can become complex, especially with diverse and intractable generators.

In this section, we introduce BIRD, Bandit-inspired generator assignment and Iterative cluster Refinement, a novel algorithm designed to iteratively learn a mixture of diverse generative models. This algorithm, inspired by the EM algorithm, is specifically tailored for complex generators with intractable likelihoods, as proposed by Banijamali et al. (2017). The key motivation is that, for most practical generators, accessing samples is generally more feasible than determining their likelihood.

Simultaneously learning clusters and generators is motivated by theoretical insights: identifying approximate clusters allows us to guarantee performance for the resulting mixture in terms of MMD. However, some datasets make it challenging to even approximate clusters. This challenge drives the creation of an algorithm that dynamically updates both clusters and generative models throughout the learning process to enhance performance.

Given a total computational budget for training generators, BIRD consists of two pivotal steps described below, distributing the computational resources between them.

1. **Cluster and Generator Initialization.** After forming initial clusters, instead of relying on traditional methods that either assign the same generator to all clusters or allocate them randomly, we introduce a bandit-inspired algorithm. This algorithm optimally pairs each cluster with a generator from a diverse set of candidates. It does so while efficiently allocating the compute budget to adaptively focus on training the parameters of the most promising generators for each cluster.

2. **Iterative Refinement of Clusters and Generators.** After bandit-inspired matching and training, to address potential errors from imperfect clustering, we propose an algorithm inspired by Banijamali et al. (2017) that iteratively refines both clusters and generators. This approach, akin to the EM algorithm, progressively enhances the identification of true data clusters. The method is outlined in Algorithm 1.

**Step 1**  **Step 2**

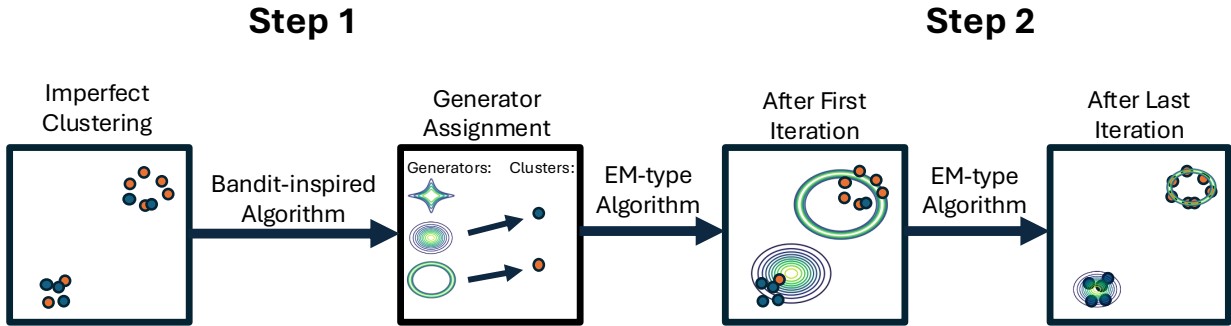

Figure 1: General workflow of our algorithm. Step 1 consists of clustering the data, which will not recover clusters perfectly in practice, and assigning generators to each cluster with our bandit-inspired algorithm. Here, there are three possible generators, and two are selected. In Step 2, we iteratively refine the clusters and the generators with an EM-type algorithm. After Step 2, we can generate synthetic data.

Figure 1 graphically represents the workflow of this BIRD algorithm. The following sections provide an in-depth explanation of each component within our approach.

### 4.1 Step 1: initialization

**Cluster Initialization.** To effectively learn a mixture of distributions, we begin by initializing the clusters. Based on the nature of the input data, we employ common clustering algorithms such as $k$-means, Uniform Manifold Approximation and Projection (UMAP), or $t$-distributed Stochastic Neighbor Embedding ($t$-SNE). The dataset is partitioned into $J$ clusters, where $J$ is determined by the chosen clustering algorithm. We conducted an additional experiment to test the sensitivity of our algorithm to the initial clustering in Appendix D.

**Generator Assignment to Clusters.** Unlike the approach taken in Banijamali et al. (2017), where generators are randomly assigned, we aim to carefully assign generators to clusters at initialization. We treat this assignment task as a stochastic multi-armed bandit problem (see Appendix I). The assignment aims to match each cluster with the most appropriate generator through sequential training. Each generator is trained using a fixed computational budget, and the less effective ones are systematically eliminated. This iterative process continues until we assign a single generator to each cluster.

Applying a bandit algorithm for generator selection is a complex task that, to our knowledge, has not been explored in the literature. To align our approach with the framework of stochastic rising bandits (Metelli et al., 2022), which is a variant of the multi-armed bandit problem where the reward distributions of the arms can change over time, we aim to select a well-behaved performance metric that improves in expectation as training progresses. Based on empirical investigations (see Appendix I) we chose the negative MMD as the reward.

Best arm identification in a stochastic rising bandit has been studied by Mussi et al. (2024), who proposed an algorithm based on Successive Rejects called R-SR Algorithm (see Algorithm 2). However, since different generators require varying computational times per iteration and bandit algorithms assume each arm selection consumes one unit of time, we cannot directly apply this algorithm with the number of iterations as the unit of time. To address this, we employ a least common multiple-based training approach, where we allocate training time in proportion to the least common multiple of the per-iteration runtimes across the candidate generators.

### 4.2 Step 2: iterative cluster and generator refinement

In this step, the algorithm iteratively estimates the posterior probabilities of the latent variables $z_i$, which indicate cluster assignment, based on the current parameter estimates. Given the intractable likelihoods of the generative models, we employ sample-based approximations.

Our goal is to estimate the membership probabilities, indicating the likelihood of each observation belonging to a specific cluster generated by a particular generator. For each generator $G_j$ (where $j = 1, \ldots, J$), we generate $l$ samples: $Y^{(j)} = (y_1^{(j)}, \ldots, y_l^{(j)})$. We then calculate a similarity score $s_{i,j}$ between each training data point $x_i$ and these samples.

Banijamali et al. (2017) originally used the mean Gaussian kernel distance for this similarity scoring. However, this measure can be ineffective in high-dimensional settings. Therefore, we adopt kernel scoring rules, specifically using the inverse of the Energy score (Gneiting & Raftery, 2007) as similarity score.

The membership probabilities are given by

$$m_{ij} \approx \mathbb{P}(x_i \text{ is generated by } G_j) = \frac{s_{ij}\pi_j}{\sum_{k=1}^{J} s_{ik}\pi_k} \ .$$

We update the mixing proportions as $\pi_j = \sum_{i=1}^{n} m_{ij}/n$.

After estimating membership probabilities, these values determine each training point's influence on learning a specific generator. Unlike the log-likelihood weighting used in EM, we propose an alternative for intractable generators. We incorporate data point importance into the mini-batch selection process for training the generators, whereas Banijamali et al. (2017) incorporated it into adjusting the learning rate. Specifically, we select mini-batches proportionally to the membership probabilities and train the generators on these

batches within a fixed training budget. The process is repeated iteratively until the maximal computational budget is reached. Algorithm 1 presents the second step in detail. A discussion about the choices for the hyperparameters can be found in Appendix B.

---

**Algorithm 1** Step 2 of the BIRD Algorithm

---

1: **Input:** Observations $x_1, \ldots, x_n$; Generators $G_i, 1 \leq i \leq J$ with $J > 0$; Initial parameters $\theta_1, \ldots, \theta_J$; Initial weights $\pi_1, \ldots, \pi_J$; Batch size $B$; Similarity measure $s$; Budget $C_{bu}$; Maximal budget $T$; $l$ number of samples generated at each iteration; $S$ number of synthetic samples.
2: **Output:** Synthetic samples $(\mathbf{x}_i^{\text{syn}})_{i=1}^S$;
3: Initialize $r \leftarrow 0$;
4: **while** $r + JC_{bu} \leq T$ **do**
5:     Generate $Y^{(j)} = (y_1^{(j)}, \ldots, y_l^{(j)})$ for $1 \leq j \leq J$;
6:     Compute similarity $s_{ij}$ between $Y^{(j)}$ and $x_i$ for $1 \leq j \leq J$ and $1 \leq i \leq n$;
7:     **for** $j = 1$ to $J$ **do**
8:         **for** $i = 1$ to $n$ **do**
9:             $m_{ij} \leftarrow s_{ij}\pi_j/(\sum_{k=1}^J s_{ik}\pi_k)$;
10:         **end for**
11:     Sample batch of size $B$ from $(x_1, \ldots, x_n)$ proportionally to $m_{ij}$;
12:     Train $G_j$ with budget $C_{bu}$ and update $\theta_j$;
13:     **end for**
14:     Update $\pi_j \leftarrow \frac{\sum_{i=1}^n m_{ij}}{n}$ for $j \in \{1, \ldots, J\}$
15:     $r \leftarrow r + JC_{bu}$;
16: **end while**
17: Generate $(\mathbf{x}_i^{\text{syn}})_{i=1}^S$ from the obtained mixture;
18: **return** $(\mathbf{x}_i^{\text{syn}})_{i=1}^S$;

---

**Remark 4.1** *While Step 2 of BIRD, which involves an EM Refinement step, is conceptually similar to approaches already proposed in the literature, such as by Banijamali et al. (2017), we introduce novel enhancements. We incorporate the Energy score as a robust similarity measure and integrate point importance through mini-batch selection, which enables compatibility with generators that employ varying training procedures.*

**Remark 4.2** *Our theoretical results do not directly apply to Algorithm 1. In our theoretical analysis, we assume the clusters are fixed and well-approximated, allowing us to use the robustness properties of the MMD to establish consistency bounds. Conversely, Algorithm 1 introduces an iterative approach to simultaneously learn the clusters and generators since it is difficult in practice to identify even approximate clusters.*

## 5 Experiments

### 5.1 Toy dataset

In this section, we illustrate visually the performance benefits of our method using a toy dataset. We generated data with the `sklearn` library (Pedregosa et al., 2011), creating the well-known two-moons dataset, which consists of two interlocking half-circle shapes.

The two-moons dataset is frequently used to evaluate clustering and classification algorithms due to its non-linear boundary. As illustrated in Figure 2a, our clustering approach, which involves first projecting the data using UMAP followed by applying the $k$-means algorithm, struggles to correctly identify and separate the two distinct clusters.

We explored two approaches for generating realistic synthetic data from the two-moons dataset. Our proposed method, the BIRD algorithm, utilizes simple neural networks tailored for moon-shaped generation as individual generators in the mixture. We compared this with a "naive mixture" method, which employs the same individual generators. In the "naive mixture", each generator is fully trained on the initial clusters.

While the "naive mixture" relies on the initial clusters, it fails to generate meaningful synthetic data if those clusters are incorrect. This limitation is evident in Figure 2c.

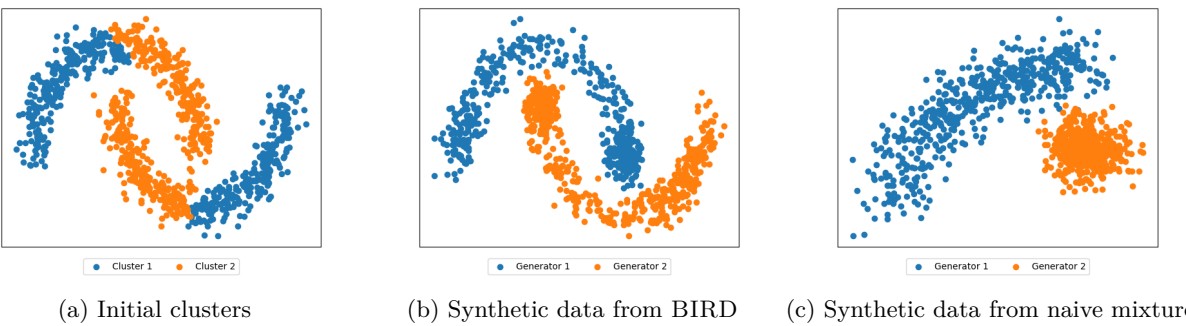

(a) Initial clusters      (b) Synthetic data from BIRD      (c) Synthetic data from naive mixture

Figure 2: Experimental results on the two-moons dataset demonstrating the effectiveness of our method compared to a naive mixture benchmark.

In contrast, Figure 2b demonstrates that the BIRD algorithm effectively captures the true clusters, producing synthetic data that reflects the two-moons shape accurately. This highlights the importance of our method's second step, which involves iterative refinement of clusters and generators in an EM-like fashion to progressively identify the underlying clusters in the data.

The two-moons example, though simplified, illustrates the motivations for our research by demonstrating the challenges in clustering complex, multi-modal datasets. It reveals the shortcomings of overly simplistic methods in combining synthetic data generators.

## 5.2 Real world datasets

**Evaluation metrics.** To evaluate the quality of synthetic data generators, we use several metrics that measure different aspects of synthetic data quality. MMD (Gretton et al., 2012) is the metric we optimize in the bandit part of the algorithm and use in our theory. Jensen-Shannon divergence (JSD) and inverse Kullback-Leibler divergence (IKL) are general measures of synthetic data quality. Probabilistic precision (P-PR; Park & Kim, 2023) measures how realistic the synthetic data points are, while probabilistic recall (P-RE) measures how much of the real distribution is covered by the synthetic distribution. The Probabilistic F1 Score (P-F1) is introduced in Appendix H.1.

The evaluation of MMD is based on the Gaussian kernel with bandwidth computed corresponding to the median heuristic (Gretton et al., 2012), while JSD and IKL are evaluated through Synthcity package (Qian et al., 2023) with all default settings. P-PR and P-RE are evaluated through the help of accompanying package provided by Park & Kim (2023).

**Available generators.** In our mixture, we have a selection of generators available: the Conditional Tabular Generative Adversarial Network (CTGAN) (Xu et al., 2019), the Tabular Variational Autoencoder (TVAE) (Xu et al., 2019), the Robust Tabular Variational Autoencoder (RTVAE) (Akrami et al., 2020), Normalizing Flows (NFlow) (Kobyzev et al., 2021), the Tabular Denoising Diffusion Probabilistic Model (TabDDPM) (Kotelnikov et al., 2023), and Adversarial Random Forests (ARF) (Watson et al., 2023).

**Competitors.** We compare our BIRD algorithm with individual generators and other methods for training mixtures of generators:

- **Naive Mixture:** Each initial cluster is assigned to the same type of generator, with clusters remaining fixed throughout.
- **Banijamali et al. (2017) Mixture:** Each initial cluster is assigned to the same type of generator, from our available options. The clusters and generators are then iteratively refined.
- **Mix UCB:** We implemented the method of Rezaei et al. (2025) as a baseline, initially dividing the budget equally to pre-train the generators, and then applying the Mix-UCB algorithm.
- **Random Mixture:** Generators are randomly assigned to each cluster instead of using a bandit-inspired approach. The second step of the BIRD algorithm is then applied.

- **Individual Generators:** The dataset is modeled using a single generator selected from the list of available generators.

Unfortunately, we couldn't fully reproduce the method of Locatello et al. (2018) due to insufficient details, particularly concerning the choice of the discriminator.

**Computational Budget.** The time budget $T$ for training the generators is set to 10 seconds. Step 1 and 2 of the BIRD algorithm are allocated 20% and 80% of the budget, respectively, based on sensitivity analysis of budget allocation choices detailed in Appendix E. In our computational budget, we focus solely on the time allocated for training generators, excluding the computation of membership probabilities, which can be expensive. While the BIRD algorithm, as well as the Mixture and Naive Mixture methods from Banijamali et al. (2017), are afforded more time than the individual generators in this setup, empirical evidence indicates that all individual generators converge within the budget given to them, so they would not benefit from the additional runtime the mixture methods are able to use. Therefore, we believe that comparing these methods remains fair. We report the performance of individual generators and Naive Mixture, selecting the training iteration with the smallest MMD. Analysis of the complexity is provided in Appendix C.

### 5.2.1 Banknote Authentication dataset

In this section, we conducted experiments on the Banknote Authentication dataset (Lohweg, 2012) from the UCI repository. This dataset includes 1372 observations from 4-dimensional data. All results are reported as the mean $\pm$ standard deviation, averaged over 10 runs.

| Generator Type | MMD (1e-3) ↓ | JSD (1e-3) ↓ | IKL (1e-1) ↑ | P-PR ↑ | P-RE ↑ | P-F1 ↑ |
|---|---|---|---|---|---|---|
| BIRD | 0.469 ± 0.186 | **3.933 ± 0.641** | **9.76341 ± 0.12130** | 0.405714 ± 0.015051 | **0.897376 ± 0.012221** | 0.5586 ± 0.0134 |
| Naive mixture | 78.835 ± 124.622 | 26.559 ± 35.060 | 8.15546 ± 1.84682 | 0.121035 ± 0.173431 | 0.538358 ± 0.321038 | 0.1959 ± 0.2381 |
| Banijamali's mixture | 7.714 ± 5.615 | 13.848 ± 14.155 | 8.91797 ± 1.33041 | 0.209458 ± 0.147522 | 0.592455 ± 0.317706 | 0.2798 ± 0.2129 |
| Mix-UCB | 11.417 ± 6.152 | 10.548 ± 0.668 | 9.38736 ± 0.23686 | 0.206793 ± 0.031956 | 0.794306 ± 0.034745 | 0.3266 ± 0.0364 |
| Random Mixture | 12.116 ± 6.863 | 19.713 ± 9.616 | 8.28383 ± 1.38414 | 0.160032 ± 0.072428 | 0.423865 ± 0.340211 | 0.2180 ± 0.1413 |
| ARF | **0.419 ± 0.052** | 4.309 ± 0.513 | 8.651744 ± 0.13326 | 0.373593 ± 0.010446 | 0.888907 ± 0.012230 | 0.5260 ± 0.0091 |
| CTGAN | 16.603 ± 1.689 | 53.16 ± 52.306 | 8.51315 ± 0.34615 | 0.147025 ± 0.017913 | 0.360716 ± 0.045909 | 0.2077 ± 0.0185 |
| NFlow | 10.124 ± 5.821 | 8.145 ± 1.355 | 9.649678 ± 0.14678 | 0.066208 ± 0.003685 | 0.784299 ± 0.032493 | 0.1221 ± 0.0063 |
| RTVAE | 13.205 ± 2.426 | 44.685 ± 1.582 | 5.996462 ± 0.10672 | 0.083014 ± 0.008969 | 0.000143 ± 0.000319 | 0.0003 ± 0.0006 |
| TabDDPM | 5.324 ± 2.25 | 6.494 ± 2.791 | 9.611834 ± 0.08279 | **0.489463 ± 0.023662** | 0.705380 ± 0.012596 | **0.5778 ± 0.0204** |
| TVAE | 13.898 ± 2.221 | 21.193 ± 1.338 | 8.48663 ± 0.08936 | 0.114446 ± 0.007894 | 0.385992 ± 0.014786 | 0.1764 ± 0.0097 |

Table 3: Experiment Results Over the Banknote Authentication Dataset. For each evaluation metric, the down (up) arrow means the lower (higher) the better respectively. The unit of value for each column is stated in the bracket right after the corresponding variable name in the table. Bold indicates the best performance. All results are reported as the mean $\pm$ std, averaged over 10 runs.

**Experimental results.** Table 3 presents our experimental findings. Within a fixed time budget, the BIRD algorithm performs the best in 3 out of 6 evaluation metrics. While TabDDPM only outperforms BIRD in the P-PR and P-F1 metrics and ARF in the MMD metric, it seems that overall BIRD provides higher quality synthetic data due to its consistent performances across all metrics.

Our method demonstrates notable improvement over other mixtures across all metrics. The superior performance of the Banijamali et al. (2017) Mixture over the Naive Mixture highlights the value of the second step in the BIRD algorithm, which involves iterative refinement of clusters and generators. Additionally, the enhanced results of our method over the Random Mixture demonstrate the advantages of using a bandit-inspired approach for assigning generators to clusters, instead of random assignment. In Appendix H.5, we perform experiments with additional baselines to provide insights into the significance of the two steps of our BIRD algorithm. Furthermore, ARF consistently outperforms other individual generators, explaining its frequent selection in our mixtures throughout experiments, see Table 4.

| Clusters | Frequency | | Generators | Frequency |
|----------|-----------|---|------------|-----------|
| 6 | 5% | | ARF | 65% |
| 7 | 10% | | RTVAE | 15% |
| 8 | 30% | | TabDDPM | 10% |
| 9 | 55% | | NFlow | 10% |

Table 4: Frequency of clusters and generator types selected by the BIRD Algorithm for the Banknote Authentication Dataset.

**Results interpretation.** In addition to analyzing performance metrics, we further evaluate the obtained mixture's characteristics. By running our algorithm multiple times, we examine the frequencies of selected cluster numbers and generator types, as shown in Table 4. The results highlight that the ARF generator is predominantly chosen, and 9 clusters are selected in the majority of runs. Furthermore, we offer a detailed analysis of a specific algorithm run, showcasing the number of clusters, the selected generators for each cluster, and the evolution of weights throughout the process; see Appendix H.2.

### 5.2.2 Wilt Dataset

The Wilt dataset from the UCI repository consists of image segments, generated by segmenting the pan-sharpened image obtained from a remote sensing study by Johnson (2013). The dataset is structured in a tabular format, comprising $4,889$ observations across five dimensions. We selected a random sample of 1000 observations. All results are reported as the mean $\pm$ standard deviation, averaged over 10 runs.

| Generator Type | MMD (1e-3) ↓ | JSD (1e-3) ↓ | IKL (1e-1) ↑ | P-PR ↑ | P-RE ↑ | P-F1 ↑ |
|----------------|--------------|--------------|--------------|--------|--------|--------|
| BIRD | **0.717 ± 0.254** | **3.152 ± 0.472** | 9.54116 ± 0.14931 | 0.784571 ± 0.007734 | **0.893646 ± 0.005541** | **0.8355 ± 0.0043** |
| Naive mixture | 195.967 ± 223.134 | 42.162 ± 37.747 | 7.23394 ± 2.62841 | 0.331965 ± 0.356911 | 0.520919 ± 0.348841 | 0.6265 ± 0.3023 |
| Banijamali's Mixture | 10.924 ± 17.764 | 13.236 ± 12.132 | 8.96860 ± 0.66291 | 0.775574 ± 0.050931 | 0.700290 ± 0.242300 | 0.7140 ± 0.1725 |
| Mix-UCB | 2.020 ± 2.242 | 5.001 ± 1.472 | **9.68704 ± 0.11048** | 0.770677 ± 0.008392 | 0.863528 ± 0.011048 | 0.8144 ± 0.0041 |
| Random mixture | 9.906 ± 4.237 | 17.533 ± 0.652 | 8.80693 ± 0.66291 | 0.810574 ± 0.074438 | 0.542776 ± 0.217401 | 0.6325 ± 0.1832 |
| ARF | 1.152 ± 0.335 | 3.814 ± 0.524 | 9.43006 ± 0.14108 | 0.757962 ± 0.011955 | 0.890924 ± 0.006112 | 0.8316 ± 0.0051 |
| CTGAN | 4.033 ± 1.730 | 17.855 ± 2.221 | 8.59009 ± 0.34408 | 0.782503 ± 0.050652 | 0.556873 ± 0.044805 | 0.6533 ± 0.0700 |
| NFlow | 2.188 ± 0.083 | 8.481 ± 0.162 | 9.51835 ± 0.06154 | 0.731015 ± 0.027852 | 0.834860 ± 0.008516 | 0.7812 ± 0.0104 |
| RTVAE | 11.014 ± 1.407 | 67.147 ± 2.180 | 5.09527 ± 0.25950 | 0.884897 ± 0.035128 | 0.000000 ± 0.000000 | 0.0000 ± 0.0000 |
| TabDDPM | 1.310 ± 0.432 | 3.863 ± 0.406 | 8.73762 ± 0.13682 | **0.811365 ± 0.021716** | 0.844694 ± 0.027302 | 0.8285 ± 0.0049 |
| TVAE | 4.205 ± 1.137 | 11.984 ± 3.745 | 9.18439 ± 0.20267 | 0.720246 ± 0.075902 | 0.676544 ± 0.073853 | 0.7057 ± 0.0221 |

Table 5: Experiment Results Over the Wilt Dataset. For each evaluation metric, the down (up) arrow denotes lower (higher) is better, respectively. The unit of value for each column is indicated in the bracket right after the corresponding variable name in the table. Bold indicates the best performance. All results are reported as the mean $\pm$ standard deviation, averaged over 10 runs.

The results presented in Table 5 show that our BIRD algorithm outperforms the other mixtures across most of the metrics. While the individual generators, ARF and TabDDPM, also perform well, they do not match the overall effectiveness of BIRD. Our BIRD algorithm demonstrates superior performance in generating high-quality synthetic data, as evidenced by its higher P-F1 score compared to other methods.

Furthermore, we present the frequencies of selected generators and the number of clusters to provide insights into the results and the obtained mixture. As shown in Table 6, ARF is the most frequently selected generator and there is a strong preference for selecting 7 clusters.

| Clusters | Frequency | Generators | Frequency |
|---|---|---|---|
| 7 | 95% | ARF | 40% |
| 5 | 2.5% | RTVAE | 25% |
| 9 | 2.5% | TabDDPM | 20% |
| | | NFlow | 15% |

Table 6: Frequency of the number of clusters and type of generator selected for the Wilt dataset.

## 6 Discussion

This study addresses the challenge of modeling complex multi-modal distributions in tabular datasets through a mixture of generative models. Given a set of generative models and a tabular dataset, our goal is to effectively combine these models to produce high-quality synthetic data. We have developed a novel algorithm called BIRD, which integrates an innovative bandit-inspired generator assignment and an iterative cluster refinement mechanism, to efficiently train mixtures of generative models. Our approach leverages diverse generators, enabling each to specialize in distinct data regions, thereby circumventing the limitations of traditional generative models like mode collapse.

Unlike the framework developed in Banijamali et al. (2017), which was limited to homogeneous generators, our approach accommodates diverse types of generators. This flexibility is made possible by our innovative application of a bandit algorithm, inspired by stochastic bandit literature, marking the first use of such a method for generative model assignment. While Step 2 of our algorithm builds on existing ideas, such as those in Banijamali's work, we have introduced several modifications to enhance performance. Notably, we have employed the inverse of the Energy score as a similarity measure between a point and a cluster. Additionally, we modified how point importance is incorporated by using it to select the batch of points used to train the generators.

Theoretically, we demonstrate that by utilizing the robustness properties of the MMD, if clusters can be approximately identified, learning an optimal mixture of generators can be achieved with an error similar to that in the well-specified case, under mild assumptions. Although our theoretical results build on those in (Chérief-Abdellatif & Alquier, 2022), we emphasize the novelty of our work in utilizing the robustness of the MMD to tackle the challenge of imperfectly identified clusters. We believe this approach paves the way for applying similar insights using various robust metrics. However, it is important to clarify that our theoretical results are formulated for simplified settings and do not serve as guarantees of convergence or correctness for the complete BIRD algorithm, where clusters and generators are iteratively updated together.

In more realistic scenarios, where clusters are challenging to approximate, we offer empirical evidence of the efficacy of BIRD algorithm, showcasing substantial improvements over simple mixture models and individual generators in generating synthetic data. This algorithm, initially developed with a focus on tabular data, is versatile and can be adapted for other data types, including images and text. This adaptation would require minor adjustments, such as selecting suitable kernels for the MMD and choosing an appropriate similarity measure. While we observed significant improvements over baseline models, opportunities remain for optimizing implementation choices like similarity measures and budget allocation, especially with high-dimensional datasets. Additionally, it is important to acknowledge that the scalability of our method is currently constrained by the computational cost of calculating membership probabilities. Despite promising outcomes from experiments on larger datasets using subsampling strategies (see Appendix H.4), developing a systematic approach to address this challenge remains an open research question and is an avenue for future research. Furthermore, extending theoretical results to robust metrics beyond MMD could be valuable.

Finally, we draw attention to our comparison with the SMOTE algorithm presented in Appendix H.6. While SMOTE demonstrates superior performance in certain examples, it is heavily reliant on the availability of class labels, unlike our BIRD algorithm, which operates label-free. Furthermore, SMOTE raises potential privacy concerns by risking the exposure of private information from the real dataset.

**Acknowledgments**

The work of A. Faul was funded by the Multidisciplinary Center for Infectious Diseases (MCID) from the University of Bern, grant number MA 14. Part of his work has been conducted during a visit at the University of Cambridge funded by the grant "Support for Young Academics" of the University of Bern. O. Räisä was supported by the Researchers Abroad (Tutkijat maailmalle)-program (project 20240109). The work of C. Tekin was supported by the Scientific and Technological Research Council of Türkiye (TUBITAK) under Grant 124E065; TUBITAK 2024 Incentive Award; by the Turkish Academy of Sciences Distinguished Young Scientist Award Program (TUBA-GEBIP-2023).

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

# A    Proof of Corollary 3.5

We consider a characteristic kernel $k$, and let $\mathcal{H}_k$ denote the reproducing kernel Hilbert space (RKHS) associated to $k$ equipped with the norm $\|\cdot\|_{\mathcal{H}_k}$. We assume that the data has been generated by the mixture distribution $\mathbb{P}_{\text{mix}} = \sum_{i=1}^{J} \lambda_i \mathbb{P}_i$ where $\mathbb{P}_1, \ldots, \mathbb{P}_J$ are probability distributions on the same space.

We denote $\tilde{\theta}_n = \{\tilde{\theta}_{n_i}^{(i)}\}_{i=1}^{J}$, the approximate parameters obtained in each identified cluster of size $n_i, \forall i \in \{1, \ldots, J\}$, as presented in Assumption 3.4 and we consider $\tilde{P}^{(n)}$ to be the empirical distribution of $n$ i.i.d. samples generated from $P_{\tilde{\theta}} = \sum_{i=1}^{J} \lambda_i \mathbb{P}_{\tilde{\theta}_{n_i}^{(i)}}$. We consider the estimated weights $\tilde{\lambda} = \{\tilde{\lambda}_i\}_{i=1}^{J}$ obtained by exactly minimizing the MMD with respect to the empirical distribution as detailed in Remark 3.1:

$$\tilde{\lambda} = (\tilde{\lambda}_1, \ldots, \tilde{\lambda}_J) \in \underset{\lambda=(\lambda_1, \ldots, \lambda_J)}{\operatorname{argmin}} \operatorname{MMD}(\tilde{P}^{(n)}, \sum_{i=1}^{J} \lambda_i \mathbb{P}_{\tilde{\theta}_{n_i}^{(i)}}) \quad \text{s.t.} \quad \lambda_i \geq 0, \sum_{i=1}^{J} \lambda_i = 1 \ .$$

By denoting $\mu_{\mathbb{P}}$ the kernel mean embedding of the distribution $\mathbb{P}$, we get that

$$\begin{aligned}
&\operatorname{MMD}_k \left( \sum_{i=1}^{J} \tilde{\lambda}_i \mathbb{P}_{\tilde{\theta}_{n_i}^{(i)}}, \mathbb{P}_{\text{mix}} \right) \\
&= \left\| \mu_{\sum_{i=1}^{J} \tilde{\lambda}_i \mathbb{P}_{\tilde{\theta}_{n_i}^{(i)}}} - \mu_{\sum_{i=1}^{J} \lambda_i \mathbb{P}_i} \right\|_{\mathcal{H}_k} \\
&= \left\| \mu_{\sum_{i=1}^{J} \tilde{\lambda}_i \mathbb{P}_{\tilde{\theta}_{n_i}^{(i)}}} - \mu_{\sum_{i=1}^{J} \lambda_i \mathbb{P}_{\tilde{\theta}_{n_i}^{(i)}}} + \mu_{\sum_{i=1}^{J} \lambda_i \mathbb{P}_{\tilde{\theta}_{n_i}^{(i)}}} - \mu_{\sum_{i=1}^{J} \lambda_i \mathbb{P}_i} \right\|_{\mathcal{H}_k} \\
&\leq \operatorname{MMD}_k \left( \sum_{i=1}^{J} \tilde{\lambda}_i \mathbb{P}_{\tilde{\theta}_{n_i}^{(i)}}, \sum_{i=1}^{J} \lambda_i \mathbb{P}_{\tilde{\theta}_{n_i}^{(i)}} \right) + \left\| \mu_{\sum_{i=1}^{J} \lambda_i \mathbb{P}_{\tilde{\theta}_{n_i}^{(i)}}} - \mu_{\sum_{i=1}^{J} \lambda_i \mathbb{P}_i} \right\|_{\mathcal{H}_k} \\
&= \operatorname{MMD}_k \left( \sum_{i=1}^{J} \tilde{\lambda}_i \mathbb{P}_{\tilde{\theta}_{n_i}^{(i)}}, \sum_{i=1}^{J} \lambda_i \mathbb{P}_{\tilde{\theta}_{n_i}^{(i)}} \right) + \left\| \sum_{i=1}^{J} \lambda_i \left( \mu_{\mathbb{P}_{\tilde{\theta}_{n_i}^{(i)}}} - \mu_{\mathbb{P}_i} \right) \right\|_{\mathcal{H}_k} \\
&\leq \operatorname{MMD}_k \left( \sum_{i=1}^{J} \tilde{\lambda}_i \mathbb{P}_{\tilde{\theta}_{n_i}^{(i)}}, \sum_{i=1}^{J} \lambda_i \mathbb{P}_{\tilde{\theta}_{n_i}^{(i)}} \right) + \sum_{i=1}^{J} \lambda_i \operatorname{MMD}_k (\mathbb{P}_{\tilde{\theta}_{n_i}^{(i)}}, \mathbb{P}_i) \ ,
\end{aligned}$$

where we used the linearity of the kernel mean embeddings and the triangular inequality. By taking the expectation with respect to the empirical samples, we get that

$$\mathbb{E}(\operatorname{MMD}_k(\sum_{i=1}^{J} \tilde{\lambda}_i \mathbb{P}_{\tilde{\theta}_{n_i}^{(i)}}, \mathbb{P}_{mix})) \leq \mathbb{E} \left[ \operatorname{MMD}_k \left( \sum_{i=1}^{J} \tilde{\lambda}_i \mathbb{P}_{\tilde{\theta}_{n_i}^{(i)}}, \sum_{i=1}^{J} \lambda_i \mathbb{P}_{\tilde{\theta}_{n_i}^{(i)}} \right) \right] + \sum_{i=1}^{J} \lambda_i \mathbb{E}(\operatorname{MMD}_k(\mathbb{P}_{\tilde{\theta}_{n_i}^{(i)}}, \mathbb{P}_i)) \ .$$

For each real cluster $i \in \{1, \ldots, J\}$ of size $n_i$, we denote $\hat{P}_i$ its empirical distribution given by

$$\hat{P}_i = \frac{1}{n_i} \sum_{x_l \in x^{*(i)}} \delta_{x_l} \ ,$$

where $\delta_{x_l}$ denotes the Dirac measure concentrated at the point $x_l$.

Let us consider a given cluster $i$ and use a method similar to the proof of Proposition 3.5 of Chérief-Abdellatif & Alquier (2022) to bound $\mathbb{E}(\operatorname{MMD}_k(\mathbb{P}_{\tilde{\theta}_{n_i}^{(i)}}, \mathbb{P}_i))$.

From Assumption 3.3, we note that the misclassification rate for cluster $i$ is less than $\epsilon_n$. We treat the sizes of both the true cluster $x^{*(i)}$ and the estimated cluster $\tilde{x}^{(i)}$ as fixed at $n_i$. Let $\mathcal{O} = x^{*(i)} \triangle \tilde{x}^{(i)}$ represent the

symmetric difference between the actual and estimated clusters $i$ and consider $\mathcal{O}_* = \mathcal{O} \cap x^{*(i)}$. Consequently, $|\mathcal{O}_*| \leq \epsilon_n \times n_i$. For any probability measure $Q$, we have that

$$
\begin{aligned}
\left| \text{MMD}_k(Q, \tilde{P}_i) - \text{MMD}_k(Q, \hat{P}_i) \right| &\leq \text{MMD}_k(\hat{P}_i, \tilde{P}_i) \\
&= \left\| \frac{1}{n_i} \sum_{j=1}^{n_i} \left( k(X_j, \cdot) - k(\tilde{X}_j, \cdot) \right) \right\|_{\mathcal{H}_k} \\
&\leq \frac{1}{n_i} \sum_{j=1}^{n_i} \left\| k(X_j, \cdot) - k(\tilde{X}_j, \cdot) \right\|_{\mathcal{H}_k} \\
&= \frac{1}{n_i} \sum_{j \in \mathcal{O}_*} \left\| k(X_j, \cdot) - k(\tilde{X}_j, \cdot) \right\|_{\mathcal{H}_k} \\
&\leq \frac{2|\mathcal{O}_*|}{n_i} \\
&\leq 2\epsilon_n \ ,
\end{aligned}
\tag{1}
$$

where we used that the kernel $k$ is bounded by 1 and that $|\mathcal{O}_*| \leq \epsilon_n \times n_i$.

We define the MMD estimator $\hat{\theta}_{n_i}^{(i)}$ such that

$$
\text{MMD}_k(\mathbb{P}_{\hat{\theta}_{n_i}^{(i)}}, \hat{P}_i) = \inf_{\theta \in \Theta_i} \text{MMD}_k(\mathbb{P}_\theta, \hat{P}_i) \ .
$$

By the triangular inequality, we have that

$$
\begin{aligned}
\text{MMD}_k(\mathbb{P}_{\tilde{\theta}_{n_i}^{(i)}}, \mathbb{P}_i) &\leq \text{MMD}_k(\mathbb{P}_{\tilde{\theta}_{n_i}^{(i)}}, \tilde{P}_i) + \text{MMD}_k(\tilde{P}_i, \mathbb{P}_i) \\
&\leq \text{MMD}_k(\mathbb{P}_{\hat{\theta}_{n_i}^{(i)}}, \tilde{P}_i) + \chi + \text{MMD}_k(\tilde{P}_i, \mathbb{P}_i) \\
&\leq \left( 2\epsilon_n + \text{MMD}_k(\mathbb{P}_{\hat{\theta}_{n_i}^{(i)}}, \hat{P}_i) \right) + \chi + \left( 2\epsilon_n + \text{MMD}_k(\hat{P}_i, \mathbb{P}_i) \right) \ ,
\end{aligned}
$$

where Assumption 3.4 is utilized to derive the second inequality with $\theta = \hat{\theta}_{n_i}^{(i)} \in \Theta_i$ and Equation 1 is applied with $Q = \mathbb{P}_{\hat{\theta}_{n_i}^{(i)}}$ and $Q = \mathbb{P}_i$ respectively, leading to the final inequality.

Consequently, we get that

$$
\begin{aligned}
\text{MMD}_k(\mathbb{P}_{\tilde{\theta}_{n_i}^{(i)}}, \mathbb{P}_i) &\leq \chi_n + 4\epsilon_n + \text{MMD}_k(\mathbb{P}_{\hat{\theta}_{n_i}^{(i)}}, \hat{P}_i) + \text{MMD}_k(\hat{P}_i, \mathbb{P}_i) \\
&\leq \chi_n + 4\epsilon_n + \text{MMD}_k(\mathbb{P}_{\theta_{0,i}}, \hat{P}_i) + \text{MMD}_k(\hat{P}_i, \mathbb{P}_i) \\
&= \chi_n + 4\epsilon_n + 2\text{MMD}_k(\hat{P}_i, \mathbb{P}_i) \ ,
\end{aligned}
$$

where the equality is obtained using Assumption 3.2. By applying the generalization bound from Theorem 3.1 of Chérief-Abdellatif & Alquier (2022) in the i.i.d. case, we obtain:

$$
\mathbb{E}(\text{MMD}_k(\mathbb{P}_{\tilde{\theta}_{n_i}^{(i)}}, \mathbb{P}_i)) \leq \chi_n + 4\epsilon_n + 4\mathbb{E}\left( \frac{1}{\sqrt{n_i}} \right) \ .
$$

Applying this result to each cluster and summing the outcomes, we find

$$
\sum_{i=1}^{J} \lambda_i \mathbb{E}(\text{MMD}_k(\mathbb{P}_{\tilde{\theta}_{n_i}^{(i)}}, \mathbb{P}_i)) \leq \chi_n + 4\epsilon_n + 4 \sum_{i=1}^{J} \lambda_i \mathbb{E}\left( \frac{1}{\sqrt{n_i}} \right) \ .
\tag{2}
$$

Furthermore, let us consider the parametric family $\mathcal{P}_\lambda$ defined by

$$
\mathcal{P}_\lambda = \left\{ P_\lambda = \sum_{i=1}^{J} \lambda_i \mathbb{P}_{\tilde{\theta}_{n_i}^{(i)}}, \lambda = (\lambda_1, \dots, \lambda_J) \text{ s.t } \lambda_i \geq 0, \sum_{i=1}^{J} \lambda_i = 1 \right\} \ .
$$

We know that the MMD estimator $\tilde{\lambda}$ is such that

$$\text{MMD}_k(P_{\tilde{\lambda}}, \tilde{P}^{(n)}) = \inf_{P_\lambda \in \mathcal{P}_\lambda} \text{MMD}_k(P_\lambda, \tilde{P}^{(n)}) ,$$

where $P_{\tilde{\lambda}} = \sum_{i=1}^{J} \tilde{\lambda}_i \mathbb{P}_{\tilde{\theta}_{n_i}^{(i)}}$. Then by the Theorem 3.1 of Chérief-Abdellatif & Alquier (2022) in the i.i.d. case we have that

$$\mathbb{E}\left[\text{MMD}_k\left(\sum_{i=1}^{J} \tilde{\lambda}_i \mathbb{P}_{\tilde{\theta}_{n_i}^{(i)}}, \sum_{i=1}^{J} \lambda_i \mathbb{P}_{\tilde{\theta}_{n_i}^{(i)}}\right)\right] \leq \frac{2}{\sqrt{n}} + \inf_{P_\lambda \in \mathcal{P}_\lambda} \text{MMD}_k(P_\lambda, \sum_{i=1}^{J} \lambda_i \mathbb{P}_{\tilde{\theta}_{n_i}^{(i)}}) .$$

Since $\sum_{i=1}^{J} \lambda_i \mathbb{P}_{\tilde{\theta}_n^{(i)}} \in \mathcal{P}_\lambda$ the infimum is zero, leading to the conclusion that

$$\mathbb{E}\left[\text{MMD}_k\left(\sum_{i=1}^{J} \tilde{\lambda}_i \mathbb{P}_{\tilde{\theta}_{n_i}^{(i)}}, \sum_{i=1}^{J} \lambda_i \mathbb{P}_{\tilde{\theta}_{n_i}^{(i)}}\right)\right] \leq \frac{2}{\sqrt{n}} . \tag{3}$$

By combining Equations 2 and 3, we get the desired result:

$$\mathbb{E}\left(\text{MMD}_k\left(\sum_{i=1}^{J} \tilde{\lambda}_i \mathbb{P}_{\tilde{\theta}_{n_i}^{(i)}}, \mathbb{P}_{mix}\right)\right) \leq \sum_{i=1}^{J} 4\lambda_i \mathbb{E}\left(\frac{1}{\sqrt{n_i}}\right) + \chi_n + 4\epsilon_n + \frac{2}{\sqrt{n}} .$$

Then by using Markov Inequality to the random variable $Z = \text{MMD}_k\left(\sum_{i=1}^{J} \tilde{\lambda}_i \mathbb{P}_{\tilde{\theta}_{n_i}^{(i)}}, \mathbb{P}_{mix}\right)$, which is almost surely positive, we have that

$$\forall \ \upsilon > 0, \ \mathbb{P}(Z \geq \upsilon) \leq \frac{\mathbb{E}(Z)}{\upsilon} .$$

If we denote $\rho = \sum_{i=1}^{J} \mathbb{E}\left(\frac{4\lambda_i}{\sqrt{n_i}}\right) + \chi_n + 4\epsilon_n + \frac{2}{\sqrt{n}}$, we have by the previous result that $\mathbb{E}(Z) \leq \rho$.

By combining the equations, we get that

$$\forall \ \upsilon > 0, \ \mathbb{P}(Z \geq \upsilon) \leq \frac{\mathbb{E}(Z)}{\upsilon} \leq \frac{\rho}{\upsilon} .$$

If we consider $\gamma = \frac{\rho}{\upsilon}$, we get that for all $0 < \gamma < 1$

$$\mathbb{P}(Z \geq \frac{\rho}{\gamma}) \leq \gamma \iff \mathbb{P}(Z \leq \frac{\rho}{\gamma}) \geq 1 - \gamma .$$

By replacing $Z$ and $\rho$ by their expressions, we obtain

$$\mathbb{P}\left(\text{MMD}_k\left(\sum_{i=1}^{J} \tilde{\lambda}_i \mathbb{P}_{\tilde{\theta}_{n_i}^{(i)}}, \mathbb{P}_{\text{mix}}\right) \leq \frac{4}{\gamma}\left(\epsilon_n + \frac{\chi_n}{4} + \frac{1}{2\sqrt{n}} + \sum_{i=1}^{J} \mathbb{E}\left(\frac{\lambda_i}{\sqrt{n_i}}\right)\right)\right) \geq 1 - \gamma .$$

## B   Implementation choices

Our algorithm involves several implementation choices to optimize performance and accuracy. These choices include:

- **Similarity Measure Selection:** One should use different similarity measures depending on the data type. For numerical data, we employ the Energy Score, which is well-suited as a proper scoring rule for measuring similarity between a point and an empirical probability distribution in a multivariate setting.

- **Computational Budget Allocation:** At each iteration of the second step of Algorithm 1, we train the generators with a budget denoted $C_{bu}$. The choice of this computational budget decides how much we want to update the generators based on the current clusters. We choose this budget based on heuristics.

Furthermore, it is essential to select an appropriate kernel for the Maximum Mean Discrepancy (MMD) in the bandit algorithm. This choice should be tailored to the specific problem at hand. We currently use the Gaussian kernel with the median heuristic for bandwidth selection, as described by Gretton et al. (2012). However, other studies, such as Dziugaite et al. (2015), suggest employing mixtures of Gaussian kernels with varying bandwidths.

Other hyperparameters, such as batch size for training generators and window size in the bandit algorithm, are currently selected based on heuristics. A more comprehensive analysis would be needed in future work to optimize these settings. Finally, users can determine the total computational budget for training generators based on their available resources. We fix this budget to 10 seconds for these experiments.

## C   Complexity of the algorithm

A detailed analysis of scaling considerations is presented below, where $J$ represents the number of clusters, $d$ the dimensionality of the observations, and $n$ the dataset size:

- **Initial Clustering**: The $K$-means algorithm operates with a complexity of $\mathcal{O}(nJd)$.

- **Bandit-Inspired Assignment**: The computational complexity of this step can be effectively managed by setting a specific budget, which should be adjusted according to the generators and dataset size.

- **Membership Probabilities Computation**: A significant challenge in scaling to larger datasets involves computing membership probabilities. By denoting $l$ as the number of elements sampled from each generator for similarity assessment, the complexity for calculating similarities between the $n$ dataset points and the $K$ clusters, using Energy scores, scales as $\mathcal{O}(J(lnd + l^2d))$. This quadratic increase in complexity relative to $l$ can pose issues. To mitigate this, we have developed various sub-sampling strategies to efficiently reduce $l$.

- **Training of Generators**: In the second phase of BIRD, we train $J$ generators using standard procedures. Training can be parallelized once the membership probabilities are established, ensuring that the requirement to train multiple generators is manageable.

Our method requires additional computational resources beyond standard single-generator training procedures due to the need for clustering and membership probability calculations. The initial clustering and membership probability computations have linear complexity in terms of $n$, $J$, and $d$. However, the complexity of computing membership probabilities scales quadratically with $l$, the number of data points considered for similarity assessment. To address this, we have developed and implemented subsampling strategies that effectively mitigate this issue. It is also possible to use another similarity metric.

Additionally, our EM-type algorithm involves multiple iterations of membership probability computations, while clustering is performed only once at the beginning. Consequently, the complexity for membership probabilities computation becomes more critical in our approach. This challenge can be particularly significant with large values of $N$, $J$, and $d$. In practice, we can mitigate this issue by reducing the frequency of membership probability computations, thereby increasing the training budget $C_{bu}$ for each generator following parameter updates.

We have conducted a time complexity analysis of our random sub-sampling approach described in Section H.4.

| Dataset Size | 2 Clusters | 5 Clusters | 10 Clusters |
|---|---|---|---|
| 500 | $2.9304 \pm 2.7472$ | $5.1733 \pm 2.4567$ | —— |
| 1000 | $4.2922 \pm 3.0883$ | $12.7463 \pm 3.1880$ | $20.7856 \pm 3.2046$ |
| 2000 | $15.8113 \pm 4.4504$ | $34.2800 \pm 5.2510$ | $64.9423 \pm 4.3418$ |

Table 7: Empirical runtime (seconds) for one iteration of Step 2 of BIRD with varying dataset size and number of clusters found by initial clustering.

Next, we examine how the computational time changes with variations in dataset size and the number of initial clusters. Using the Wilt dataset, we report the computational time for one iteration of Step 2 in the BIRD algorithm across different values for these parameters.

The results indicate that the computational complexity aligns with the scalability analysis. By taking into account the variation in computational time over different runs, as shown by the standard deviation, it seems reasonable that complexity increases linearly with the number of clusters and dataset size. It should be noted that results with 10 clusters and 500 points were not obtained because some clusters contained too few points to allow effective training.

## D   Sensitivity to initial clustering

We conducted an additional experiment to test the sensitivity of our algorithm to the initial clustering. For this purpose, we manually deteriorate the quality of the initial clustering by randomly flipping a proportion of points between clusters and see how the performance of the BIRD algorithm changes. We performed experiments with 25%, 50% and 75% of points flipped into other clusters, we considered the Wilt dataset with the same budget and hyperparameters as in the original experiments (see Section 5.2). The results presented in the Table 8 show that the performance decreases with poor initial clustering. In terms of Probabilistic Precision (P-PR) and Probabilistic Recall (P-RE), this decrease is not dramatic (metrics results are within $\pm 2\%$).

| Random Proportion | MMD (1e-3) $\downarrow$ | JSD (1e-3) $\downarrow$ | IKL (1e-1) $\uparrow$ | P-PR $\uparrow$ | P-RE $\uparrow$ |
|---|---|---|---|---|---|
| 25% | 0.532 | 2.618 | 9.68048 | 0.787081 | 0.894134 |
| 50% | 0.547 | 3.065 | 9.57840 | 0.788960 | 0.889261 |
| 75% | 0.646 | 3.842 | 9.56200 | 0.778305 | 0.901704 |

Table 8: Sensitivity analysis for the proportion of points flipped from initial clusters. For each evaluation metric, the down (up) arrow means the lower (higher) the better respectively. The unit of value for MMD and JSD is scaled by $10^{-3}$, and IKL by $10^{-1}$.

## E   Sensitivity to budget allocation

To study the effect of the budget allocation between the steps of our algorithm, we ran our method with various allocations to assess how this parameter influences performance. We denote by $\alpha$ the hyperparameter corresponding to the budget allocated to Step 1 and we ran experiments on the Wilt dataset with all hyperparameters the same to the original experiments described in Section 5.2, except the budget proportion $\alpha$. We provide here an overview of the obtained results for some selected values of $\alpha$ for the BIRD algorithm.

The results from Table 9 show that the best performances in terms of MMD and JSD are attained for $\alpha = 0.2$. In terms of Probabilistic Precision and Probabilistic Recall, it seems that performance is approximately at the same level while $\alpha$ is between 0.1 and 0.5, and drop heavily when $\alpha$ becomes higher than 0.5. Overall, choosing a value of $\alpha = 0.2$ as in the main paper, is a reasonable choice. Hence, this value is used for the

| $\alpha$ | MMD (1e-3) $\downarrow$ | JSD (1e-3) $\downarrow$ | IKL (1e-1) $\uparrow$ | P-PR $\uparrow$ | P-RE $\uparrow$ |
|---|---|---|---|---|---|
| 0.1 | 0.586 | 2.882 | 9.71532 | 0.786700 | 0.891381 |
| 0.2 | 0.129 | 2.565 | 9.54787 | 0.776410 | 0.895885 |
| 0.5 | 0.781 | 4.045 | 9.26192 | 0.770004 | 0.881362 |
| 0.9 | 1.207 | 14.301 | 8.92574 | 0.727702 | 0.808941 |

Table 9: Overview of the obtained results for selected values of $\alpha$ for the BIRD algorithm. For each evaluation metric, the down (up) arrow means the lower (higher) the better respectively. The unit of value for MMD and JSD is scaled by $10^{-3}$, and IKL by $10^{-1}$.

experiments in this paper. It should be noted that these results were obtained following a single run of the BIRD algorithm.

## F    Robustness of similarity measures

In the second step of our BIRD algorithm, we utilize the inverse of the Energy Score as a similarity measure, which is a proper scoring rule (Gneiting & Raftery, 2007). In contrast, Banijamali et al. (2017) paper employs the mean Gaussian kernel distance between a point and a cluster. We demonstrate the robustness of the Energy Score using a simple simulated example, highlighting its advantages over the mean Gaussian distance in our context.

We conducted a simulation involving two distinct clusters. The first cluster was generated from a Gaussian distribution with a mean vector of $\mu_1 = [0,0]^\mathsf{T}$ and an identity covariance matrix. The second cluster was generated similarly, but with a mean vector of $\mu_2 = [2,2]^\mathsf{T}$. Additionally, we introduced two outlier points into the first cluster. To analyze the clustering, we calculated the Energy Scores and mean Gaussian kernel distances with bandwidth $\sigma = 1$ from a reference point at $x^* = [0,0]^\mathsf{T}$ to both clusters. This reference point is located at the center of the first cluster, and should then be closer to the first cluster, which is reflected in a lower Energy Score. However, the mean Gaussian kernel distance to the second cluster is surprisingly smaller. This phenomenon is illustrated in Figure 3 and is due to the lack of robustness of the mean Gaussian kernel distance.

## G    Weak clustering structure of real datasets

The theoretical section assumes a meaningful clustering structure for deriving a consistency bound. However, our BIRD algorithm is versatile and can be applied to any tabular dataset without specific assumptions, although it is expected to perform better with clearly clustered data.

In this section, we demonstrate that the real datasets used in this paper lack a clear clustering structure. We applied the UMAP method for dimension reduction to visualize the datasets in two dimensions, conducting this analysis for all four datasets. The plots in Figure 4 generally indicate a lack of distinct clustering. For the Banknote Authentication dataset (Figure 4a) and the Credit Card Fraud Detection dataset (Figure 4b), there appears to be some structure, but the number of clusters is unclear. In contrast, the MiniBooNE_PID and Wilt datasets show no apparent clustering structure.

We also examined the Silhouette scores (Shahapure & Nicholas, 2020) obtained using the $K$-means clustering method for various values of $k$ in Figure 5. The Silhouette score, ranging from $-1$ to 1, reflects the quality of clustering and the degree of separation between clusters.

Figures 5c and 5d show that the Wilt and MiniBooNE_PID datasets exhibit poorer clustering quality compared to the other datasets, as evidenced by their lower Silhouette scores. Notably, the scores for these datasets do not exceed 0.5.

In Figure 6, we illustrate a dataset created using the *make_blobs* function in Python, featuring 3 overlapping blobs. Applying the $K$-means clustering algorithm with 3 clusters yields a Silhouette score of 0.48, indicating

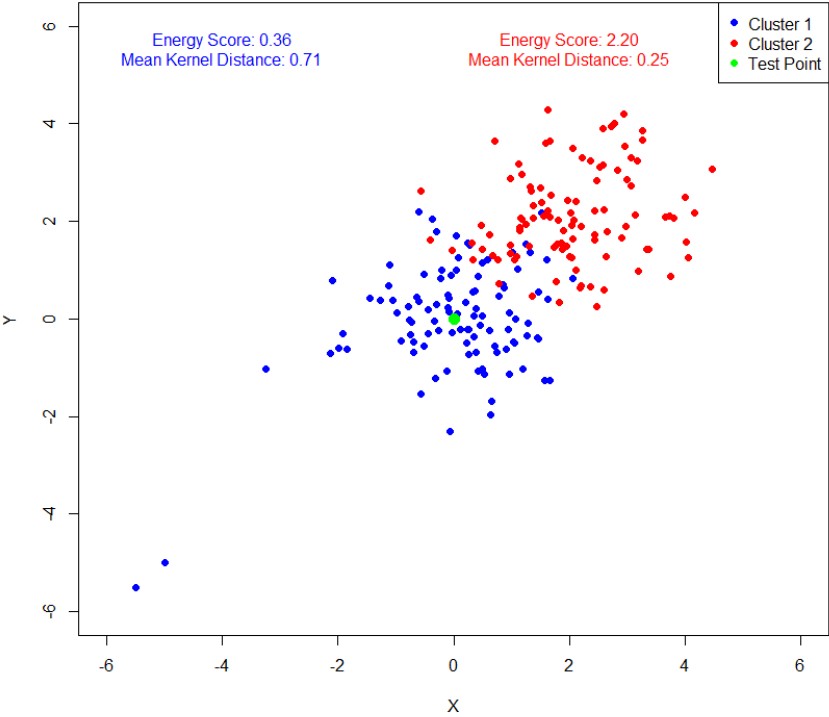

Figure 3: Energy scores and mean Gaussian kernel distances of the test point (green) to the first (blue) and second (red) clusters.

a scenario with discernible clustering structures despite overlapping clusters, which is similar to the Silhouette scores obtained for both the MiniBooNE_PID and Wilt datasets.

In conclusion, both the MiniBooNE_PID and Wilt datasets exhibit weak or nonexistent clustering structures. Nevertheless, our BIRD algorithm performs well on these datasets, demonstrating that BIRD does not require a clear clustering structure to be effective.

## H    Experimental details

The Banknote Authentication dataset (Section 5.2) is first shuffled, then randomly divided into two parts: 20% for Step 1 and 80% for Step 2. Within Step 1, 2 seconds are assigned for bandit selection of generators if applicable. Table 10 illustrates the computational capacity utilized in our experiments.

| Generator Type | ARF | CTGAN | TabDDPM | NFlow | RTVAE | TVAE |
|---|---|---|---|---|---|---|
| Iteration Numbers | $1204 \pm 17$ | $112 \pm 11$ | $865 \pm 22$ | $304 \pm 12$ | $92 \pm 2$ | $94 \pm 1$ |

Table 10: Resulting numbers of iterations for Baseline Competitors on Banknote Authentication dataset with computation budget of 10 seconds, and the computation device of Apple Silicon M4Pro with 18GB memory. All results are reported as the mean $\pm$ std, averaged over 10 runs.

All experiments have been performed in *Python 3.11.5*. The hyperparameters of the individual generators have been set to their default values in the *arfpy* package for the ARF generator and the *Synthcity* (Qian et al., 2023) for the other generators. This includes for example learning rates, batch sizes and weight decays. In

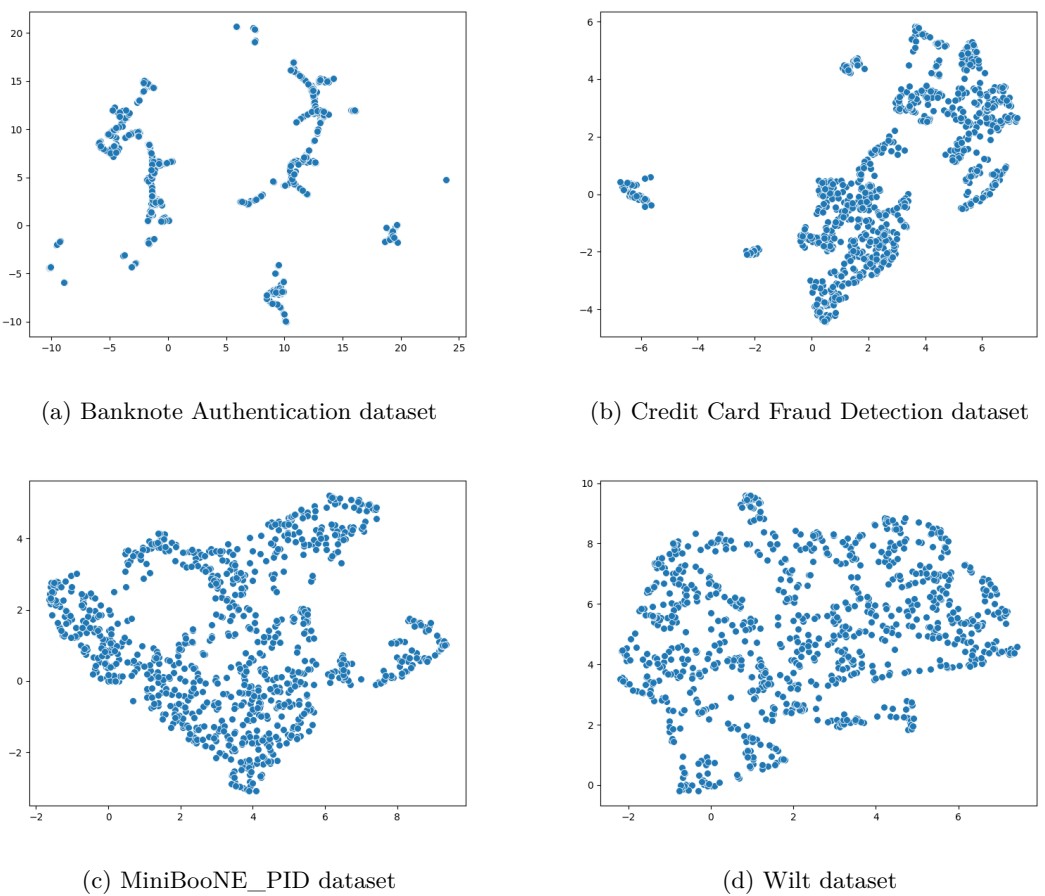

(a) Banknote Authentication dataset

(b) Credit Card Fraud Detection dataset

(c) MiniBooNE_PID dataset

(d) Wilt dataset

Figure 4: Visualization of the real-datasets used in the paper after dimension reduction with UMAP.

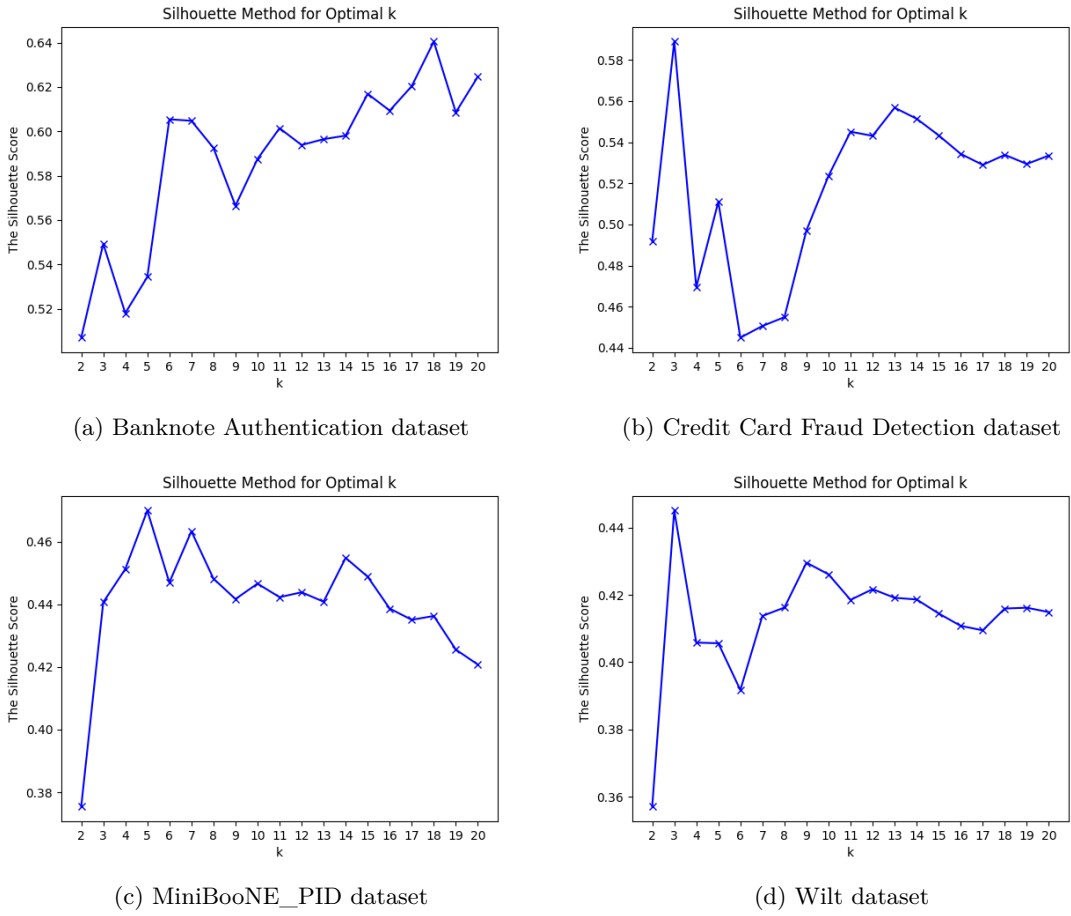

(a) Banknote Authentication dataset

(b) Credit Card Fraud Detection dataset

(c) MiniBooNE_PID dataset

(d) Wilt dataset

Figure 5: Silhouette score for $K$-means clustering method with different values of $K$ for the various datasets.

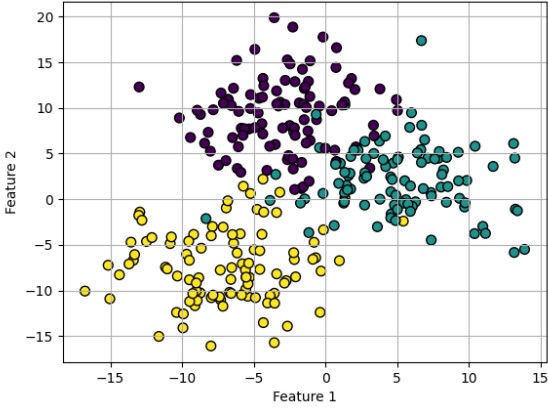

Figure 6: Visualization of a dataset with 3 overlapping blobs.

particular, for all the generators based on neural networks, it corresponds to Kaiming Initialization: Uniform for weights and uniform for bias.

As the goal of the method is simultaneously learning the parameters of the mixtures components as well as the clusters, we aim to be able to escape from a poor initialization of the individual generators.

The initial clustering step involves two main parts. First, the selected dataset is projected into a two-dimensional space using the UMAP method with default settings from the *UMAP* package. Next, the data is clustered using the $K$-means algorithm from the *sklearn* package, also utilizing default settings. To determine the optimal number of clusters, $K$, we evaluated values ranging from 2 to 10 using the Silhouette score (Shahapure & Nicholas, 2020).

For the bandit-inspired algorithm, we used the R-SR algorithm (see Appendix I), and allocated 50 time units for training all generators, approximately equating to 2 seconds in total. This time unit is based on the generator with the shortest average training time, determined by evaluating all generators over 100 iterations.

The Maximum Mean Discrepancy (MMD) is calculated using a Gaussian kernel, with the bandwidth determined by the median heuristic as described in Section B. For estimation, we generate 100 samples from the generators and apply the U-estimator introduced in Lemma 6 of Gretton et al. (2012).

In Step 2, to compute membership probabilities, we generate $l = 1,000$ samples from the generators using their current parameter values. The Energy score is then calculated using the *scoringRules* package in *R*.

For the baseline competitors, specifically the individual generators, the entire dataset is utilized for both training and evaluation.

For the other competitors, which use mixtures of generators, the dataset is divided in the same way as in the experiments with the BIRD algorithm: 20% for the initial clustering step and 80% for the remainder of the algorithm.

Both the *Naive Mixture* and the mixture inspired by Banijamali et al. (2017) randomly assign one generator to every initial cluster. In contrast, the *Random Mixture* assigns potentially different generators to each cluster at random.

Furthermore, while both the *Random Mixture* and the Banijamali et al. (2017) inspired mixture include a second step of iterative cluster refinement using membership probabilities, the *Naive Mixture* directly trains each generator with the entire computational budget on the initial clusters without further cluster refinement.

## H.1 Metrics

The evaluation of MMD is based on the Gaussian kernel with bandwidth computed corresponding to the median heuristic (Gretton et al., 2012), while JSD and IKL are evaluated through Synthcity package (Qian et al., 2023) with all default settings. P-PR and P-RE are evaluated through the accompanying package provided by the authors of Park & Kim (2023).

The Probabilistic F1 score (P-F1) corresponds to the harmonic mean of the Probabilistic Precision and the Probabilistic Recall.

$$P\text{-}F1 = 2 \times \frac{P\text{-}PR \times P\text{-}RE}{P\text{-}PR + P\text{-}RE} \;.$$

## H.2 Additional experiments

### H.2.1 Credit Card Fraud Detection Dataset

In this section, we present experimental results for synthetic data generation similarly to Section 5.2. We consider the Credit Card Fraud Detection Dataset which contains information about transactions made by credit cards in September 2013 by European cardholders. It contains data about a total of $284,807$ transactions and only numerical input variables which are the result of a Principal Component Analysis

(PCA) transformation. The dataset is publicly available at `https://www.kaggle.com/datasets/mlg-ulb/creditcardfraud`.

The experiments summarized in Table 11 involve a fixed sample of 1,000 randomly selected data points from the Credit Card Fraud Detection Dataset. A total of 20 seconds is allocated for these experiments due to the high number of features and the extended time required for full convergence.

| Generator Type | MMD (1e-3) ↓ | JSD (1e-3) ↓ | IKL (1e-1) ↑ | P-PR ↑ | P-RE ↑ | P-F1 ↑ |
|---|---|---|---|---|---|---|
| BIRD | **1.906 ± 1.515** | **6.378 ± 2.929** | 9.27328 ± 0.06391 | 0.468566 ± 0.078628 | 0.795691 ± 0.058072 | 0.5846 ± 0.0468 |
| Naive Mixture | 197.489 ± 254.740 | 45.892 ± 42.667 | 6.07838 ± 3.11481 | 0.174958 ± 0.250850 | 0.623146 ± 0.412858 | 0.2029 ± 0.2206 |
| Banijamali's mixture | 25.138 ± 21.339 | 19.213 ± 12.523 | 7.99611 ± 1.51338 | 0.534003 ± 0.232009 | 0.605491 ± 0.243739 | 0.4989 ± 0.1012 |
| Mix-UCB | 101.822 ± 27.842 | 33.213 ± 4.988 | 4.197 ± 0.4513 | 0.396667 ± 0.042069 | 0.934609 ± 0.018138 | 0.5556 ± 0.0376 |
| Random Mixture | 19.923 ± 10.638 | 23.233 ± 12.083 | 8.54704 ± 0.36309 | 0.597679 ± 0.251271 | 0.412036 ± 0.050313 | 0.4659 ± 0.0980 |
| ARF | 2.482 ± 1.292 | 6.419 ± 1.336 | 8.99981 ± 0.29777 | 0.314598 ± 0.020789 | **0.815830 ± 0.023858** | 0.4535 ± 0.0187 |
| CTGAN | 37.053 ± 7.445 | 16.115 ± 3.342 | 8.85289 ± 0.20707 | 0.579759 ± 0.023151 | 0.512022 ± 0.019602 | 0.5432 ± 0.0049 |
| NFlow | 6.745 ± 3.080 | 12.358 ± 6.808 | **9.49701 ± 0.20378** | 0.304914 ± 0.045414 | 0.680647 ± 0.036210 | 0.4188 ± 0.0400 |
| RTVAE | 32.121 ± 12.163 | 66.857 ± 18.839 | 5.39402 ± 0.49663 | **0.836923 ± 0.023157** | 0.266681 ± 0.042040 | 0.4045 ± 0.0500 |
| TabDDPM | 38.537 ± 8.976 | 19.746 ± 11.548 | 7.83914 ± 0.24907 | 0.520137 ± 0.024947 | 0.754111 ± 0.029245 | **0.6150 ± 0.0150** |
| TVAE | 27.003 ± 0.011697 | 23.635 ± 9.877 | 8.87221 ± 0.23482 | 0.652175 ± 0.077301 | 0.432904 ± 0.050928 | 0.5175 ± 0.0426 |

Table 11: Combined experimental results over the Credit Card Fraud Detection Dataset. For each evaluation metric, the down (up) arrow means the lower (higher) the better respectively. Bold indicates the best performance. All results are reported as the mean ± std, averaged over 5 runs.

The results presented in Table 11 highlight the performance of our BIRD algorithm compared to other mixture models across all considered metrics. In particular, it performs 23% better than the second-best model, ARF, in terms of MMD. We observe that while other mixtures perform poorly in every metric—even compared to individual generators—BIRD achieves the best overall performance in terms of MMD and JSD. Although the normalizing flow generator (NFlow) performs better in IKL, it falls short compared to BIRD in all other metrics. Note that the other mixture methods perform poorly compared to BIRD.

In terms of P-F1 score, which combines P-PR and P-RE, TabDDPM is the best method followed by our BIRD algorithm.

We can conclude that BIRD significantly outperforms all other mixtures and demonstrates overall superior performance compared to individual generators.

We present the frequencies of selected generators and the number of clusters to provide deeper insights into the BIRD algorithm. As shown in Table 12, ARF and RTVAE are the only generators selected by our method, with a marked preference for ARF. Additionally, the most frequently observed number of clusters is 3.

| Clusters | Frequency | Generators | Frequency |
|---|---|---|---|
| 3 | 60% | ARF | 70% |
| 6 | 25% | RTVAE | 30% |
| 9 | 15% | | |

Table 12: Frequency of the number of clusters and type of generator selected for the Credit Card Fraud Detection Dataset.

### H.2.2 Wilt dataset

The Wilt dataset from the UCI repository consists of image segments, generated by segmenting the pan-sharpened image obtained from a remote sensing study by Johnson (2013). The dataset is structured in a tabular format, comprising $4,889$ observations across five dimensions. We selected a random sample of 1000 observations. The results are provided in the Table 5.

Additionally, we provide details about the number of clusters, selected generators per cluster and the weights for the obtained mixture in Table 13, for a particular run of the algorithm to illustrate how we can analyze the resulting mixture. We can see that the ARF has been selected for most of the clusters and that the weights evolve during the training process.

| | |
|---|---|
| **Number of Clusters** | 7 |
| **Selected Generators by the Bandit Algorithm** | ARF 1, ARF 2, ARF 3, ARF 4, ARF 5, RTVAE 1, ARF 6 |
| **Initial Weights** | 0.0913, 0.1432, 0.2076, 0.1489, 0.1168, 0.1690, 0.1231 |
| **Final Weights** | 0.1200, 0.1833, 0.1751, 0.1170, 0.1934, 0.1044, 0.1066 |

Table 13: Experimental details for a single run of the BIRD algorithm.

### H.3 Capturing the distribution of minority classes

In this section, we examine the capability of our mixture algorithm to accurately capture the distribution of minority classes. This is a crucial factor in developing a fair and effective synthetic data generator.

### H.3.1 Credit Card Fraud Detection dataset

We selected a subset of the Credit Card Fraud Detection dataset with 1000 observations and 10 features such that the positive class (frauds) account for 2% of the observations while it corresponds only to 0.172% of all transactions in the original dataset.

We trained every individual generator presented in Section 5.2 and our BIRD algorithm on this dataset. We compare the synthetic data obtained with the different methods and the real data using Probabilistic Precision (P-PR) and Probabilistic Recall (P-RE) by splitting both the real and synthetic data into two parts corresponding respectively into majority class and minority class samples. The results of this experiment can be found in Table 1 of the Introduction.

We can see that the BIRD algorithm is performing better than the other generators in both the majority and the minority classes. Most of the generators, except BIRD and ARF, are performing very poorly especially in the minority class.

These results show that using a mixture of generative models allows to capture the distribution underrepresented classes in complex datasets. This is important for fairness considerations.

### H.3.2 Simulated data

We used the *make blobs* function from the *Python* package *sklearn* to generate data with two different clusters. The dataset contains 1000 observations with 10 features, one cluster contains 97% of the points and the other cluster contains 3% of the points. We performed dimension reduction with Principal Components Analysis (PCA) to reduce the number of dimensions to 2 and be able to visualize the results in Figure 7. In Figure 7, we see that a single complex generator, Adversarial Random Forests (ARF) (Watson et al., 2023), has difficulty efficiently capturing the small cluster on the right side of the plot. In contrast, a mixture of two ARFs successfully captures both clusters.

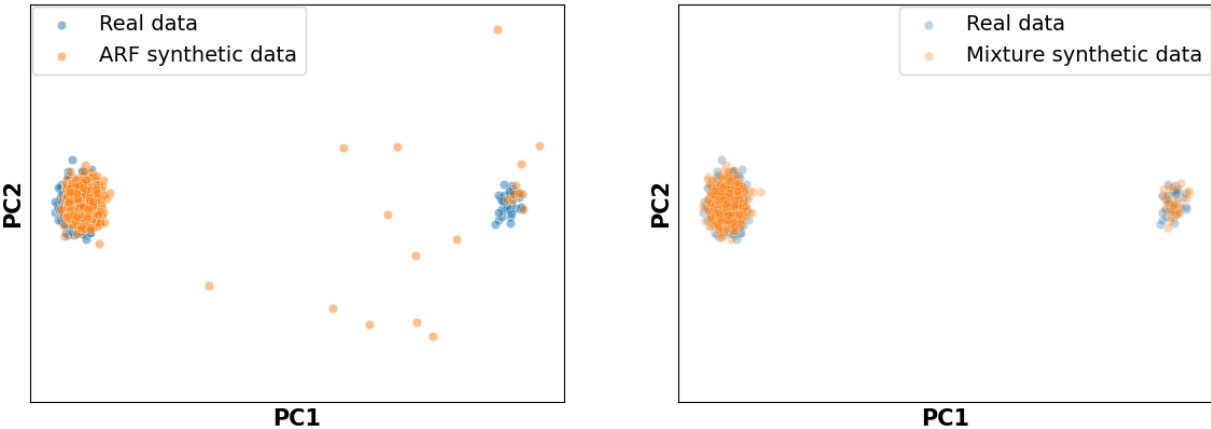

Figure 7: Synthetic samples from ARF and a Mixture of ARFs.

Similarly to above, we generated data with different synthetic data generators and we evaluated the quality of the synthetic datasets with the harmonic mean of P-PR and P-RE denoted by P-F1 in both on the minority and majority clusters. The results presented in Table 14 show that BIRD algorithm outperform the other synthetic generators in the terms of P-F1 score in the minority class and performs well in correctly identifying the proportion of points in each class. However, the ARF and TVAE outperforms BIRD in the majority class. This shows the benefit of using mixtures for correctly capturing the distribution of underrepresented classes.

|  | BIRD | ARF | CTGAN | TabDDPM | NFlow | TVAE |
|---|---|---|---|---|---|---|
| Prop.* (%) | 97.8 | 98.4 | 98.9 | 95.6 | 97.2 | 99.7 |
| P-F1 Majority ↑ | 0.7881 | 0.7950 | 0.0191 | 0.7706 | 0.6705 | **0.8566** |
| P-F1 Minority↑ | **0.7715** | 0.4943 | 0 | 0 | 0 | † |

Table 14: Performance of synthetic data generators evaluated by P-F1 scores for majority and minority classes. *True proportion is 97%. † indicates insufficient data for metric estimation. Bold values indicate the best performance.

### H.4 Experiments on large datasets

For the other experiments in this paper, the dataset size was constrained by the computational time needed to compute membership probabilities, limiting us to $1,000$ points. In this section, we conducted experiments on larger datasets by randomly sub-sampling points during similarity calculations, which significantly reduced computational time.

First, we used the MiniBooNE_PID dataset (Roe, 2005) with $20,000$ data points. The results presented in Table 15 show that our BIRD algorithm performs significantly better than the other methods in terms of MMD, JSD and IKL.

The sub-sampling process is carried out in the following manner: during each iteration of Step 2, we randomly select $2,000$ data points from the total $20,000$. We then calculate their membership probabilities. This approach significantly reduces the RAM usage while preserving the algorithm's overall efficiency.

We also conducted an experiment on the California Housing dataset, which includes 20,640 observations, and 10 numerical variables. In this experiment, we selected $10,000$ points from the dataset and every numerical variables. We applied random sub-sampling during the computation of similarity scores by selecting $1,000$ points. The results are presented in the Table 16, and the dataset is publicly available at `https://www.kaggle.com/datasets/camnugent/california-housing-prices`.

| Generator Type | MMD (1e-3) ↓ | JSD (1e-3) ↓ | IKL (1e-1) ↑ | P-PR ↑ | P-RE ↑ |
|---|---|---|---|---|---|
| BIRD | **2.980 ± 0.794** | **0.999 ± 0.174** | **9.22886 ± 0.27653** | 0.409803 ± 0.006164 | 0.903449 ± 0.013041 |
| Naive mixture | 288.994 ± 245.058 | 39.905 ± 37.145 | 4.57110 ± 3.17187 | 0.191489 ± 0.273209 | 0.691338 ± 0.345877 |
| Banijamali's mixture | 241.169 ± 253.435 | 32.22 ± 33.903 | 5.36801 ± 3.61730 | 0.278167 ± 0.311445 | 0.595166 ± 0.464903 |
| Mix-UCB | 24.814 ± 5.654 | 6.666 ± 1.159 | 8.49603 ± 0.16144 | 0.367286 ± 0.014062 | **0.992846 ± 0.002402** |
| Random mixture | 43.664 ± 68.025 | 4.273 ± 2.364 | 7.49510 ± 2.10787 | 0.381265 ± 0.235827 | 0.863253 ± 0.145128 |
| ARF | 6.031 ± 2.261 | 1.012 ± 0.185 | 9.17215 ± 0.11263 | 0.357982 ± 0.046912 | 0.907358 ± 0.011522 |
| CTGAN | 46.22 ± 8.753 | 2.815 ± 0.243 | 9.13999 ± 0.52489 | 0.149513 ± 0.045038 | 0.905282 ± 0.043921 |
| NFlow | 72.09 ± 27.618 | 6.608 ± 1.964 | 6.07824 ± 1.05561 | 0.213437 ± 0.010271 | 0.907882 ± 0.005221 |
| RTVAE | 37.090 ± 16.942 | 81.723 ± 1.025 | 6.07174 ± 0.53275 | **0.729125 ± 0.014641** | 0.064946 ± 0.046071 |
| TabDDPM | 18.403 ± 0.012 | 63.686 ± 0.396 | 7.49154 ± 0.01632 | 0.195206 ± 0.007152 | **0.953307 ± 0.025488** |
| TVAE | 15.296 ± 5.615 | 2.153 ± 0.513 | 8.99508 ± 0.02085 | 0.582362 ± 0.071160 | 0.479192 ± 0.040327 |

Table 15: Experiment Results Over the MiniBooNE_PID Dataset. For each evaluation metric, the down (up) arrow denotes lower (higher) is better, respectively. The unit of value for each column is indicated in the bracket right after the corresponding variable name in the table. Bold indicates the best performance. All results are reported as the mean ± standard deviation, averaged over 10 runs.

| Generator Type | MMD (1e-3) ↓ | JSD (1e-3) ↓ | IKL (1e-1) ↑ | P-PR ↑ | P-RE ↑ | P-F1 ↑ |
|---|---|---|---|---|---|---|
| BIRD | **0.732 ± 0.311** | 3.969 ± 0.988 | 9.51615 ± 0.09091 | 0.756180 ± 0.007163 | 0.851598 ± 0.030500 | 0.800748 ± 0.010656 |
| Naive mixture | 1.984 ± 2.331 | 7.418 ± 5.423 | 9.56459 ± 0.32685 | 0.734279 ± 0.123947 | 0.665922 ± 0.300051 | 0.641307 ± 0.255153 |
| Banijamali's mixture | 5.986 ± 8.924 | 8.132 ± 5.783 | 9.47501 ± 0.35374 | 0.700321 ± 0.093396 | 0.678512 ± 0.311698 | 0.642030 ± 0.265209 |
| Mix-UCB | 6.015 ± 3.438 | 6.367 ± 0.738 | 9.68052 ± 0.08522 | 0.711817 ± 0.011951 | 0.850313 ± 0.013211 | 0.774770 ± 0.002656 |
| Random mixture | 3.233 ± 0.765 | 5.516 ± 1.165 | **9.76961 ± 0.04666** | 0.664034 ± 0.026055 | 0.794563 ± 0.012478 | 0.723079 ± 0.012985 |
| ARF | 0.906 ± 0.289 | **3.810 ± 0.311** | 9.40359 ± 0.08014 | 0.737311 ± 0.003097 | 0.876962 ± 0.004459 | **0.801085 ± 0.001782** |
| CTGAN | 61.913 ± 48.303 | 19.785 ± 5.029 | 8.59752 ± 0.37913 | 0.434579 ± 0.275731 | 0.601659 ± 0.095186 | 0.474664 ± 0.241999 |
| NFlow | 34.718 ± 3.243 | 12.384 ± 1.730 | 8.91865 ± 0.27728 | 0.211252 ± 0.048852 | **0.898809 ± 0.013358** | 0.339530 ± 0.063687 |
| RTVAE | 5.282 ± 0.706 | 20.032 ± 3.583 | 8.40210 ± 0.38136 | **0.780445 ± 0.021229** | 0.096442 ± 0.118939 | 0.150913 ± 0.181217 |
| TabDDPM | 14.947 ± 5.625 | 13.921 ± 2.526 | 7.34605 ± 0.47994 | 0.676249 ± 0.031996 | 0.824484 ± 0.017597 | 0.742605 ± 0.019726 |
| TVAE | 8.410 ± 0.888 | 13.202 ± 0.375 | 9.18640 ± 0.04375 | 0.758621 ± 0.035116 | 0.563958 ± 0.080904 | 0.642564 ± 0.039950 |

Table 16: Experiment Results Over the California Housing Prices Dataset. For each evaluation metric, the down (up) arrow denotes lower (higher) is better, respectively. The unit of value for each column is indicated in the bracket right after the corresponding variable name in the table. Bold indicates the best performance. All results are reported as the mean ± standard deviation, averaged over 10 runs.

The results in Table 16 demonstrate that BIRD outperforms other generators in terms of MMD, while also maintaining competitive P-F1, JSD and IKL scores. This highlights BIRD as the most effective generator for this problem.

## H.5 Additional baselines

In this section, we introduce additional baselines to provide insights into the significance of the two steps in our BIRD algorithm.

Firstly, as the comparison between BIRD's Step 2 and the analogous Step 2 from Banijamali et al. is not explicitly addressed in the main paper, we included competitors using a random single-class generator for

all clusters, followed by BIRD's Step 2. Additionally, we consider a baseline that executes Step 1 of BIRD followed by Step 2 from Banijamali et al.

Moreover, we include a competitor similar to our Random Mixture approach, but with two modifications: (1) a randomly chosen generator for each cluster, and (2) Step 2 from Banijamali et al. To address variability in model class quality, which can result in large errors for some mixture models, we evaluate competitors where a single generator is selected for each cluster, followed by either the Banijamali et al. Step 2 or BIRD Step 2, or by simply training the generators naively.

Table 23 displays the performance of all these methods in comparison to BIRD and Banijamali's methods on the Wilt dataset.

The results presented in Table 23 indicate that individual generators generally perform better when using BIRD's second step compared to Banijamali's. This suggests the superiority of BIRD's Step 2. Notably, employing a naive training approach for Step 2 usually leads to suboptimal performance. Furthermore, the competitor employing the same random generator for all clusters combined with BIRD's Step 2 outperforms Banijamali's mixture in most metrics, confirming that BIRD's Step 2 is more efficient than Banijamali's Step 2.

Overall, BIRD remains the best-performing method, indicating that none of the competitors using a single generator for all clusters outperforms it, which shows the importance of mixing generative models.

Lastly, it is important to note that certain generators, like CTGAN or TabDDPM, show particularly poor performance especially when trained naively for Step 2, as evidenced by P-PR and P-RE scores of 0. This is likely due to the limited computational budget for training, which prevents some generators from reaching convergence, resulting in poor outcomes. However, it's noteworthy that TabDDPM, when trained with either BIRD's Step 2 or Banijamali's Step 2, achieves satisfactory outcomes, particularly excelling in the P-F1 score.

## H.6  Comparison to SMOTE algorithm

The SMOTE algorithm (Pradipta et al., 2021) generates samples from minority classes and assumes the presence of a variable in the dataset indicating the class of each observation. In contrast, our BIRD algorithm can generate data from any dataset without requiring class labels. To compare our algorithm with SMOTE, we used the Wilt dataset, which contains class information. We generated data using the SMOTE implementation from the *imbalanced-learn* package and conducted hyperparameter optimization with the *optuna* package in Python. We generated 100 simulated datasets and present the results in the table below.

| Generator Type | MMD (1e-3) ↓ | JSD (1e-3) ↓ | IKL (1e-1) ↑ | P-PR ↑ | P-RE ↑ |
|---|---|---|---|---|---|
| **BIRD** | $0.717 \pm 0.254$ | $3.152 \pm 0.472$ | $9.54 \pm 0.149$ | $0.784 \pm 0.0077$ | $0.8936 \pm 0.005$ |
| **SMOTE** | $1.358 \pm 0.0756$ | $8.572 \pm 0.284$ | $9.770 \pm 0.098$ | $0.957 \pm 0.0025$ | $0.9372 \pm 0.0005$ |

The similarity measures in the table indicate that synthetic datasets generated using the SMOTE algorithm are more closely aligned with real datasets than those generated by BIRD for several metrics, including IKL, P-PR, and P-RE. However, because SMOTE relies on nearest neighbors, it poses privacy concerns by potentially exposing private information from the real dataset. This issue is highlighted in the recent paper by Ganev et al. (2026). SMOTE is likely to reveal private information from the real dataset, which is a crucial concern in many fields such as medicine.

## H.7  Removing top-performing generators

In this section, we aim to study the algorithm's behavior and especially the generators to clusters assignment when the top-performing generators, such as ARF, are removed. We have conducted an additional experiment to explore this. Table 17 shows the frequency at which each generator is selected by the bandit algorithm for the Banknote Authentication dataset, when only ARF is missing. We notice that TabDDPM is then mostly

| Clusters | Frequency | Generators | Frequency |
|----------|-----------|------------|-----------|
| 3 | 90% | TabDDPM | 90% |
| 6 | 5% | NFlow | 5% |
| 9 | 5% | RTVAE | 5% |

Table 17: Frequency of the number of clusters and type of generator selected for the Banknote Authentication dataset without ARF only.

| Clusters | Frequency | Generators | Frequency |
|----------|-----------|------------|-----------|
| 3 | 90% | NFlow | 50% |
| 6 | 5% | TVAE | 40% |
| 9 | 5% | RTVAE | 10% |

Table 18: Frequency of the number of clusters and type of generator selected for the Banknote Authentication dataset without ARF and TabDDPM.

selected, so we removed it and ran the bandit algorithm again when both ARF and TabDDPM are removed from the list of generators, and we notice that in Table 18 that several generators are often chosen.

Additionally, we ran the full algorithm excluding ARF and TabDDPM from the list of generators and compared the results to the baselines. Table 19 demonstrates a significant decline in performance compared to the original results in Table 3 of the main paper. Despite this, BIRD remains the top performer in terms of MMD and P-F1 scores. However, NFlow performs the best in terms of JSD and IKL.

| Generator Type | MMD (1e-3) ↓ | JSD (1e-3) ↓ | IKL (1e-1) ↑ | P-PR ↑ | P-RE ↑ | P-F1 ↑ |
|----------------|--------------|--------------|--------------|--------|--------|--------|
| BIRD | **9.587 ± 1.159** | 17.747 ± 1.371 | 8.94972 ± 0.11765 | **0.140516 ± 0.009723** | 0.465669 ± 0.015969 | **0.215629 ± 0.010488** |
| Naive mixture | 117.260 ± 141.339 | 36.181 ± 40.897 | 7.73831 ± 2.18291 | 0.024233 ± 0.017443 | 0.417923 ± 0.288081 | 0.060707 ± 0.013675 |
| Banijamali's mixture | 11.071 ± 2.731 | 18.820 ± 15.325 | 8.57727 ± 1.55891 | 0.115376 ± 0.026687 | 0.415916 ± 0.265740 | 0.147819 ± 0.083541 |
| Mix-UCB | 54.934 ± 10.492 | 31.181 ± 8.202 | 7.66439 ± 0.40323 | 0.125040 ± 0.061944 | 0.123557 ± 0.130439 | 0.088983 ± 0.060471 |
| Random mixture | 16.654 ± 4.105 | 25.418 ± 7.925 | 7.51094 ± 1.26152 | 0.118058 ± 0.062279 | 0.191215 ± 0.167033 | 0.132628 ± 0.112368 |
| CTGAN | 21.895 ± 12.562 | 54.966 ± 50.917 | 8.33875 ± 0.42357 | 0.139314 ± 0.014989 | 0.325276 ± 0.102973 | 0.191407 ± 0.031981 |
| NFlow | 14.072 ± 4.608 | **8.026 ± 1.206** | **9.67579 ± 0.11756** | 0.069021 ± 0.004169 | **0.791199 ± 0.026395** | 0.126914 ± 0.006871 |
| RTVAE | 13.951 ± 2.719 | 44.639 ± 1.615 | 6.16086 ± 0.35692 | 0.073490 ± 0.020426 | 0.000410 ± 0.000603 | 0.000800 ± 0.001171 |
| TVAE | 13.764 ± 2.218 | 21.050 ± 1.334 | 8.53013 ± 0.12747 | 0.112067 ± 0.008002 | 0.387082 ± 0.013284 | 0.173730 ± 0.010401 |

Table 19: Experiment Results Over the Banknote Dataset when ARF and TabDDPM are removed from the list of generators. All results are reported as the mean ± std, averaged over 10 runs.

# I  Best arm identification in stochastic rising bandits

## I.1  Stochastic bandits

A stochastic multi-armed bandit game is parameterized by the number of arms $J$, the number of rounds (or budget) $T$, and $J$ probability distributions $\nu_1, \ldots, \nu_J$ associated respectively with arm 1 to arm $J$. These distributions are unknown to the player. For $t = 1, \ldots, T$, at round $t$, the player selects an arm $K_t \in [J]$, and observes a reward drawn from $\nu_{K_t}$ independently from past actions and outcomes. At the end of the $T$ rounds, the player chooses an arm, denoted $K_T$, which is evaluated in terms of the difference between the

mean reward of the optimal arm and the mean reward of $K_T$. If we let $\mu_1, \ldots, \mu_J$ be the respective means of $\nu_1, \ldots, \nu_J$ and $\mu^* = \max_{k \in [J]} \mu_k$, then the simple regret of the learner is $r_T = \mu^* - \mu_{K_T}$.

We assume that the support of $\nu_k$, $k \in [J]$ is in $[0, 1]$ and that there is a unique optimal arm $i^*$ such that $\mu_{i^*} = \mu^*$. We define the suboptimality gap of arm $i \neq i^*$ as $\Delta_i = \mu^* - \mu_i$. While the minimum nonzero gap is considered to be $\Delta_{min} = \min_{i \neq i^*} \Delta_i$. The error probability is given as $e_T = \mathbb{P}(K_T \neq i^*)$.

## I.2 Stochastic rising bandits

In the typical stochastic bandit setting described above, the reward distributions for each arm are fixed at $\mu_1, \ldots, \mu_J$, hence we are in a stationary setting. Next, we consider the possibility of these distributions changing over time. Specifically, we examine a rested bandit scenario (Tekin & Liu, 2012) where the expected reward of an arm increases whenever it is pulled.

The player selects an arm $K_t \in [J]$, plays it, and observes a reward $x_t \sim \nu_{K_t}(N_{K_t,t})$, where $\nu_{K_t}(N_{K_t,t})$ is the reward distribution of arm $K_t$ at round $t$ and depends on the number of pulls performed so far. In particular, the number of pulls of arm $i$ by the end of $t$ rounds is:

$$N_{i,t} = \sum_{\tau=1}^{t} \mathbb{I}\{K_\tau = i\} .$$

The rewards are stochastic, formally $x_t := \mu_{K_t}(N_{K_t,t}) + \eta_t$, where $\mu_{K_t}(\cdot)$ is the expected reward of arm $K_t$ and $\eta_t$ is a zero-mean $\sigma^2$-subgaussian noise, conditioned to the past. We assume that the expected rewards $\mu_i(t)$ are bounded in $[0, 1]$, $\forall i \in \{1, \ldots, J\}, \forall t \in \{1, \ldots, T\}$.

The objective is to select the arm with the highest expected reward with a high probability, given a fixed budget $T$. Unlike the stationary best arm identification problem, where the optimal arm remains constant, this scenario requires us to determine when to evaluate an arm's optimality. Optimality is defined by considering the arm with the highest expected reward at the end of $T$ pulls. It's important to note that the optimal arm may change depending on the value of $T$.

We assume that the expected rewards satisfy the following assumptions (Mussi et al., 2024), which seems numerically verified for our generators by taking the negative MMD as reward, as shown in Figure 8.

**Assumption 1** (Non-decreasing and concave expected rewards).

*Let $\nu$ be a rested multi-arm bandit , defining $\gamma_i(n) := \mu_i(n+1) - \mu_i(n)$, for every $n \in [1, T]$ and every arm $i \in [J]$ the expected rewards are non-decreasing and concave:*

- *Non-decreasing:   $\gamma_i(n) \geq 0$,*

- *Concave:   $\gamma_i(n+1) \leq \gamma_i(n)$.*

**Assumption 2** (Polynomial $\gamma_i(n)$)

*There exists $c > 0$ and $\beta > 1$ such that for every arm $i$ and number of pulls $n$, it holds that $\gamma_i(n) \leq cn^{-\beta}$.*

At round $t \leq T$, we would like to estimate the mean reward of each arm at the end of the budget $T$ i.e., we are looking for estimators of $\mu_i(T)$, $1 \leq i \leq J$.

Let $\epsilon \in (0, \frac{1}{2})$. We use an adaptive, arm-dependent window size $h(N_{i,t}) = \lceil \epsilon N_{i,t-1} \rceil$ to include only the most recent samples collected from arm $i$. At this stage, we might also consider employing exponential smoothing as an alternative to the sliding window method.

From this, we can define two estimators of $\mu_i(T)$ similarly to Metelli et al. (2022):

- A pessimistic estimator assumes that the mean of the reward distribution remains constant. It calculates the average of the recent $h(N_{i,t-1})$ rewards collected from the $i$th arm. The estimator is

defined by

$$\hat{\mu}_i(N_{i,t-1}) = \frac{1}{h(N_{i,t-1})} \sum_{\tau=N_{i,t-1}-h(N_{i,t})+1}^{N_{i,t-1}} x_\tau \ . \tag{4}$$

- An optimistic estimator assumes that the mean of the reward distribution increases linearly with each selection of an arm. This estimator is developed by enhancing the pessimistic estimator, $\hat{\mu}_i(N_{i,t-1})$, with an estimate of the expected increment that occurs in the subsequent steps up to $T$, and is given by

$$\breve{\mu}_i(N_{i,t-1}) = \hat{\mu}_i(N_{i,t-1}) + \sum_{\tau=N_{i,t-1}-h(N_{i,t})+1}^{N_{i,t-1}} (T-\tau)\frac{x_\tau - x_{\tau-h(N_{i,t-1})}}{h(N_{i,t-1})^2} \ . \tag{5}$$

## I.3   Identify the optimal generator

We want to apply the framework of best arm identification in stochastic rising bandits to selecting the optimal generator for synthesizing a given dataset.

Assume we need to synthesize a dataset $X = (x_1, \ldots, x_n)$ and have access to a set of generators $\{G_\theta^{(1)}, \ldots, G_\theta^{(J)}\}$, where each $G_\theta^{(i)}$ is a parametric generative model, for $i = 1, \ldots, J$. Our goal is to identify which generator is best suited for synthesizing $X$.

Training each generator individually and selecting the best-performing one based on a specific criterion can be time-intensive, especially if the number of potential generators is large.

To address this, we propose using a sequential procedure to identify the best generator. Each generator corresponds to an arm, and each action involves applying training steps to an arm. We define the stochastic reward at time $t$ for arm $i$ as the negative MMD between samples generated by $G_{\theta_t}^{(i)}$, using current parameters $\theta_t$ and the original sample $X$.

We show empirically in Section I.4 that this reward is increasing with respect to the number of iterations. This trend is anticipated, as the generators typically improve on average with additional iterations. Moreover, experimental results suggest the concavity of this reward, which supports the use of algorithms designed for stochastic rising bandits.

Using the estimators described in Section I.2, Mussi et al. (2024) proposed two algorithms: one based on the Upper Confidence Bound (UCB) and the other on the Successive Rejects (SR) method.

We present adaptations of two algorithms to address the problem of identifying optimal generators.

Our R-UCBE Algorithm, see Algorithm 3, utilizes the value

$$\breve{\beta}_{I_t}^T(N_{I_t,t}, \alpha) = \sigma(t - N_{i,t-1} + h(N_{i,t-1}) - 1)\sqrt{\frac{\alpha}{h(N_{i,t-1})^3}}$$

which guides the algorithm's exploration. Selecting the appropriate exploration parameter can be challenging, as it requires an understanding of the problem's complexity—an often elusive factor.

Conversely, our adaptation of the R-SR Algorithm, see Algorithm 2, eliminates the need for an exploration parameter altogether.

Since different generators have varying computational times per iteration, directly using iterations as a budget is impractical. To overcome this, we employ a least common multiple-based training approach, where we allocate training time in proportion to the least common multiple of the per-iteration runtimes across the candidate generators.

In our experiments, we primarily used the R-SR Algorithm 2 because of its superior performance.

## I.4   Empirical validation of the hypothesis

In this section, we empirically validate the assumptions of stochastic rising bandits outlined in Section I.2. Specifically, we demonstrate that the negative Maximum Mean Discrepancy (MMD), which we selected as

---

**Algorithm 2** Implementation of R-SR Bandit

---

1: **Input:** Time Budget $T$, List of Available Generators $[J]$, List of Clusters $[C]$.
2: **for** $c \in [C]$ **do**
3:     Initialize $t \leftarrow 1$, $N_0 = 0$, $\mathcal{X}_0 = [J]$
4:     **for** $j \in [J-1]$ **do**
5:         Update $N_j := \lceil \frac{1}{\overline{log}(J)} \frac{T-J}{J+1-j} \rceil$, where $\overline{log}(J) := \frac{1}{2} + \sum_{i=2}^{J} \frac{1}{i}$
6:         **for** $i \in \mathcal{X}_{j-1}$ **do**
7:             **for** $t \in [N_{j-1}+1, \ N_j]$ **do**
8:                 Choose generator $i$ and train the generator with data points from cluster $c$ for a unit of time, equivalently $n$ iterations
9:                 Trained generator $i$ generates a set of samples $S$
10:                Observe reward $x_t = -\text{MMD}(c, S)$
11:             **end for**
12:             Update $\hat{\mu}_i(N_j) = \frac{1}{N_j - N_{j-1}} \sum_{t=N_{j-1}+1}^{N_j} x_t$
13:         **end for**
14:         Define $\bar{I}_j \in \arg\min_{i \in \mathcal{X}_{j-1}} \hat{\mu}_i(N_j)$
15:         Update $\mathcal{X}_j = \mathcal{X}_{j-1} \setminus \{\bar{I}_j\}$
16:     **end for**
17:     **Recommend** A unique generator $\hat{I}_c^*(T) \in \mathcal{X}_{J-1}$.
18: **end for**
19: Return the optimal selection of generators $[\hat{I}_c^*(T)]$

---

**Algorithm 3** Implementation of R-UCBE Bandit

---

1: **Input:** Time Budget $T$, List of Available Generators $[J]$, List of Clusters $[C]$, Window Size $\epsilon$, Exploration Rate $\alpha$.
2: **for** $c \in [C]$ **do**
3:     Initialize $N_{i,0} = 0$, $B_i^T(0) = +\infty$, $\forall i \in [J]$.
4:     **for** $t \in [T]$ **do**
5:         Compute $I_t \in argmax_{i \in [J]} B_i^T(N_{i,t-1})$
6:         Choose generator $I_t$ and train the generator with data points from cluster $c$ for a unit of time, equivalently $n$ iterations
7:         Trained generator $I_t$ generates a set of samples $S$
8:         Observe reward $x_t = -\text{MMD}(c, S)$
9:         $N_{I_t,t} = N_{I_t,t-1} + 1$
10:        $N_{i,t} = N_{i,t-1}, \forall i \neq I_t$
11:        $h(N_{i,t}) = \lceil \epsilon N_{i,t-1} \rceil$
12:        Update $B_{I_t}^T(N_{I_t,t}) = \breve{\mu}_{I_t}^T(N_{I_t,t}) + \breve{\beta}_{I_t}^T(N_{I_t,t})$
13:     **end for**
14:     **Recommend** $\hat{I}_c^*(T) \in \arg\max_{i \in [J]} B_i^T(N_{i,T})$
15: **end for**
16: Return the optimal selection of generators $[\hat{I}_c^*(T)]$

---

our reward function, is a non-decreasing, concave function of the number of training iterations. To achieve this, we trained each generator in our mixture using the Banknote Authentication dataset (Lohweg, 2012). After each training iteration, we generated $m$ samples from the generator and computed the negative MMD relative to the training data.

Figure 8a illustrates the evolution of the negative MMD during the training of a CTGAN on the Banknote Authentication dataset. It appears that, on average, the assumptions of non-decreasing behavior and concavity are verified. Similarly, Figure 8b demonstrates that these assumptions are also validated on average for the other generators. Please note that the negative MMD we utilize in this study does not align with the

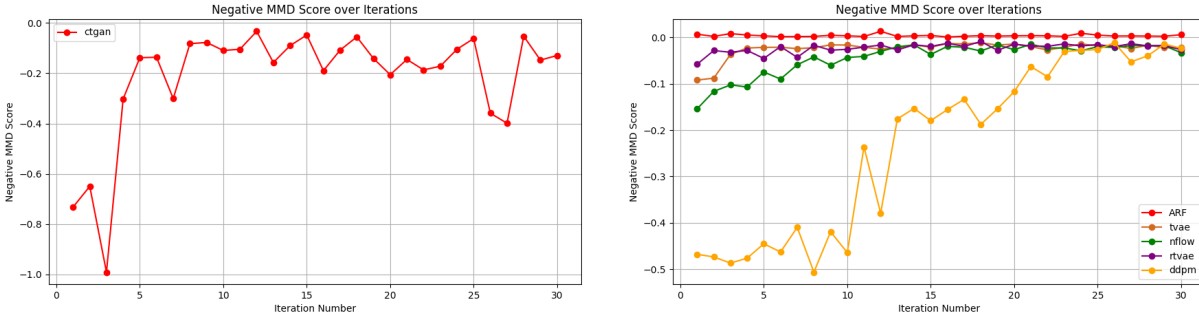

(a) Negative MMD over training for the CTGAN     (b) Negative MMD over training for the other generators

Figure 8: Experiment results over the Banknote Authentication Dataset.

theoretical framework proposed by Mussi et al. (2024), as the expected rewards are not confined to the $[0, 1]$ range.

## J   Sensitivity to the number of clusters

We conduct experiments using simulated data, progressively increasing the number of clusters while maintaining the same value range. We then compare the performance of the BIRD algorithm for different number of clusters in the Table 20. From these results, it appears that across various metrics, the performance of BIRD remains relatively stable as the number of clusters increases.

| Number of clusters | MMD (1e-3) ↓ | JSD (1e-3) ↓ | IKL (1e-1) ↑ | P-PR ↑ | P-RE ↑ | P-F1 ↑ |
|---|---|---|---|---|---|---|
| 2 CLUSTERS | $0.835 \pm 0.408$ | $5.003 \pm 1.149$ | $7.86016 \pm 0.52757$ | $0.728912 \pm 0.020023$ | $0.822336 \pm 0.014957$ | $0.772508 \pm 0.006426$ |
| 3 CLUSTERS | $0.542 \pm 0.206$ | $4.601 \pm 0.959$ | $8.67949 \pm 0.14629$ | $0.617682 \pm 0.020022$ | $0.861116 \pm 0.012114$ | $0.719193 \pm 0.014498$ |
| 5 CLUSTERS | $0.782 \pm 0.340$ | $4.611 \pm 0.861$ | $9.03061 \pm 0.28702$ | $0.611636 \pm 0.013376$ | $0.818655 \pm 0.019167$ | $0.699956 \pm 0.008266$ |
| 10 CLUSTERS | $0.705 \pm 0.338$ | $4.616 \pm 0.526$ | $8.74804 \pm 0.38476$ | $0.542465 \pm 0.017046$ | $0.919598 \pm 0.016565$ | $0.682119 \pm 0.010226$ |

Table 20: Performances of the BIRD algorithm with different number of clusters in the data.

## K   Alternative similarity measures

We investigated modifying the similarity measure itself. We conducted an experiment on the Wilt dataset, implementing the BIRD algorithm with alternative distances, such as Gaussian and Euclidean distances, in place of the Energy score. While these alternatives are quicker to compute and significantly reduce the computational burden, they lead to a decline in algorithm performance compared to the Energy score. This can be seen in the Table 21. Note that we omit standard deviations in this table, since we have performed only one repetition of this experiment.

## L   Sensitivity to random sub-sampling during membership probabilities computation.

As highlighted in the main paper, particularly in the discussion, Section 6, the primary computational bottleneck of the BIRD algorithm lies in calculating membership probabilities. In Section K, we examine different similarity measures to address this challenge. However, while these measures are quicker to compute, they result in decreased performance.

To scale our method for larger datasets, we explored the use of random sub-sampling during the computation of membership probabilities, as detailed in Section H.4. Specifically, after generating $l$ points with given

| Similarity measure | MMD | JSD | IKL | P-PR | P-RE | Time for a similarity computation (sec) |
|---|---|---|---|---|---|---|
| Gaussian Distance | 0.006383 | 0.021676 | 0.784856 | 0.837465 | 0.067049 | 0.0001 |
| Euclidean Distance | 0.018894 | 0.015844 | 0.867694 | 0.948619 | 0.159254 | 0.0005 |
| Energy Score | 0.001566 | 0.004997 | 0.952334 | 0.805953 | 0.856252 | 0.0011 |

Table 21: Performances and computational time for the BIRD algorithm on the Wilt dataset with different similarity measures.

parameter values, only a random proportion is used for computing membership probabilities. In that section, the sampling proportion was set to 10%. Here, we investigate various proportions and assess the trade-off between computational efficiency and performance loss.

Using the Wilt dataset, we applied the BIRD algorithm with different proportions of randomly sampled points for membership probability computations. We report both performance metrics and computational time for a single iteration of Step 2 of BIRD in Table 22.

| Sub-sampling Proportion | MMD (1e-3) ↓ | JSD (1e-3) ↓ | IKL (1e-1) ↑ | P-PR ↑ | P-RE ↑ | P-F1 ↑ | Time per iteration of Step 2 |
|---|---|---|---|---|---|---|---|
| 10% | $3.788 \pm 0.929$ | $5.018 \pm 0.721$ | $9.35486 \pm 0.10259$ | $0.711067 \pm 0.012433$ | $\mathbf{0.898276 \pm 0.009393}$ | $0.793670 \pm 0.005109$ | $\mathbf{0.7710 \pm 0.1762}$ |
| 30% | $3.254 \pm 1.133$ | $4.192 \pm 0.541$ | $9.43534 \pm 0.12710$ | $0.711201 \pm 0.020759$ | $0.891764 \pm 0.014986$ | $0.791196 \pm 0.016184$ | $1.1629 \pm 0.2287$ |
| 50% | $3.155 \pm 2.297$ | $4.639 \pm 1.690$ | $9.40403 \pm 0.16875$ | $0.741046 \pm 0.014076$ | $0.867651 \pm 0.011912$ | $0.799197 \pm 0.003423$ | $1.7501 \pm 0.1892$ |
| 100% | $\mathbf{0.717 \pm 0.254}$ | $\mathbf{3.152 \pm 0.472}$ | $\mathbf{9.54116 \pm 0.14931}$ | $\mathbf{0.784571 \pm 0.007734}$ | $0.893646 \pm 0.005541$ | $\mathbf{0.8355 \pm 0.0043}$ | $4.1338 \pm 0.4761$ |

Table 22: Experiment Results Over the Wilt Dataset with different random sub-sampling proportions during membership probabilities computation. For each evaluation metric, the down (up) arrow denotes lower (higher) is better, respectively. The unit of value for each column is indicated in the bracket right after the corresponding variable name in the table. Bold indicates the best performance. All results are reported as the mean ± standard deviation, averaged over 10 runs. The computational time per iteration of Step 2 of BIRD is also reported.

Table 22 indicates that reducing the proportion of sub-sampled points for computing membership probabilities generally results in decreased performance across most metrics, while computational time is reduced. This suggests a trade-off between performance loss and computational efficiency with lower sub-sampling proportions. For instance, when the sub-sampling proportion is reduced from 50% to 10%, the average MMD increases by 20.06% and JSD by 8.1%, while computational time decreases by 55.9%. Interestingly, the Probabilistic Recall (P-RE) metric does not follow this trend. This anomaly may result from the generator sampling more tightly around the true distribution, though P-PR decreases.

| Generator Type | MMD (1e-3) ↓ | JSD (1e-3) ↓ | IKL (1e-1) ↑ | P-PR ↑ | P-RE ↑ | P-F1 ↑ |
|---|---|---|---|---|---|---|
| BIRD | 0.717 ± 0.254 | 3.152 ± 0.472 | 9.54116 ± 0.14931 | 0.784571 ± 0.007734 | 0.893646 ± 0.005541 | 0.8355 ± 0.0043 |
| Banijamali's mixture | 10.924 ± 17.764 | 13.236 ± 12.132 | 8.96860 ± 0.66291 | 0.775574 ± 0.050931 | 0.700290 ± 0.242300 | 0.7140 ± 0.1725 |
| (1) BIRD step 1 (2) Banijamali step 2 | 3.269 ± 2.530 | 4.733 ± 1.539 | 9.61651 ± 0.11038 | 0.738062 ± 0.049304 | 0.891696 ± 0.026441 | 0.806193 ± 0.021255 |
| (1) random generator (2) Banijamali step 2 | 4.392 ± 1.753 | 7.976 ± 1.278 | 9.43625 ± 0.20743 | 0.770782 ± 0.034682 | 0.787729 ± 0.033791 | 0.778166 ± 0.015561 |
| (1) random generator (2) BIRD step 2 | 7.0704 ± 7.833 | 12.386 ± 11.729 | 9.15825 ± 0.65423 | 0.800092 ± 0.062551 | 0.675659 ± 0.268428 | 0.687477 ± 0.214723 |
| (1) ARF (2) train naively | 1.235 ± 0.278 | 3.312 ± 0.848 | 9.54998 ± 0.13331 | 0.766615 ± 0.006858 | 0.888859 ± 0.009417 | 0.823168 ± 0.002677 |
| (1) CTGAN (2) train naively | 726.828 ± 52.070 | 95.110 ± 8.164 | 2.85923 ± 0.29850 | 0.000000 ± 0.000000 | 0.000000 ± 0.000000 | 0.000000 ± 0.000000 |
| (1) NFlow (2) train naively | 261.103 ± 15.730 | 36.668 ± 0.897 | 5.47203 ± 0.20195 | 0.012387 ± 0.003321 | 0.935870 ± 0.006721 | 0.024432 ± 0.006487 |
| (1) RTVAE (2) train naively | 69.596 ± 12.686 | 15.686 ± 2.188 | 8.84870 ± 0.29084 | 0.587675 ± 0.048265 | 0.404314 ± 0.028145 | 0.478144 ± 0.027497 |
| (1) TabDDPM (2) train naively | 590.339 ± 2.147 | 83.518 ± 6.069 | 2.74624 ± 0.09964 | 0.000000 ± 0.000000 | 0.000000 ± 0.000000 | 0.000000 ± 0.000000 |
| (1) TVAE (2) train naively | 462.627 ± 46.979 | 24.710 ± 5.452 | 8.05140 ± 0.26656 | 0.366274 ± 0.067018 | 0.193002 ± 0.065339 | 0.248925 ± 0.066925 |
| (1) ARF (2) BIRD step 2 | 0.946 ± 0.460 | 3.189 ± 0.479 | 9.59659 ± 0.12725 | 0.783279 ± 0.014779 | 0.889165 ± 0.011250 | 0.832709 ± 0.003681 |
| (1) CTGAN (2) BIRD step 2 | 16.531 ± 12.296 | 11.606 ± 1.818 | 9.32515 ± 0.14898 | 0.771970 ± 0.060409 | 0.684015 ± 0.028188 | 0.724118 ± 0.027634 |
| (1) NFlow (2) BIRD step 2 | 3.198 ± 1.675 | 3.924 ± 0.927 | 9.65084 ± 0.13749 | 0.704660 ± 0.026780 | 0.863799 ± 0.011370 | 0.775792 ± 0.012378 |
| (1) RTVAE (2) BIRD step 2 | 13.713 ± 3.386 | 36.039 ± 1.229 | 7.79996 ± 0.24993 | 0.852420 ± 0.020262 | 0.137307 ± 0.051240 | 0.233451 ± 0.076939 |
| (1) TabDDPM (2) BIRD step 2 | 1.105 ± 0.445 | 4.245 ± 0.506 | 9.36019 ± 0.18189 | 0.817069 ± 0.018785 | 0.870423 ± 0.021774 | 0.842723 ± 0.014695 |
| (1) TVAE (2) BIRD step 2 | 6.930 ± 1.336 | 15.312 ± 1.520 | 9.21679 ± 0.10252 | 0.871157 ± 0.014019 | 0.609243 ± 0.041429 | 0.716070 ± 0.024252 |
| (1) ARF (2) Banijamali's step 2 | 1.065 ± 0.771 | 3.768 ± 0.560 | 9.51836 ± 0.22447 | 0.786056 ± 0.013280 | 0.881740 ± 0.012934 | 0.830989 ± 0.002307 |
| (1) CTGAN (2) Banijamali's step 2 | 38.573 ± 30.584 | 15.223 ± 3.864 | 8.90634 ± 0.26523 | 0.810382 ± 0.036379 | 0.632102 ± 0.064915 | 0.707294 ± 0.032217 |
| (1) NFlow (2) Banijamali's step 2 | 5.230 ± 3.517 | 4.861 ± 0.925 | 9.70298 ± 0.12134 | 0.712221 ± 0.027266 | 0.857866 ± 0.015727 | 0.777830 ± 0.011849 |
| (1) RTVAE (2) Banijamali's step 2 | 9.616 ± 3.892 | 40.537 ± 6.200 | 7.14964 ± 0.52827 | 0.817788 ± 0.060087 | 0.088538 ± 0.065247 | 0.152719 ± 0.107133 |
| (1) TabDDPM (2) Banijamali's step 2 | 1.917 ± 1.652 | 4.152 ± 0.435 | 9.44696 ± 0.06811 | 0.821360 ± 0.020484 | 0.866707 ± 0.008158 | 0.843247 ± 0.008649 |
| (1) TVAE (2) Banijamali's step 2 | 6.401 ± 2.378 | 15.587 ± 1.115 | 9.16987 ± 0.14041 | 0.864397 ± 0.017678 | 0.628090 ± 0.021936 | 0.727256 ± 0.014539 |

Table 23: Experiment Results Over the Wilt Dataset. For each evaluation metric, the down (up) arrow denotes lower (higher) is better, respectively. The unit of value for each column is indicated in the bracket right after the corresponding variable name in the table. All results are reported as the mean ± standard deviation, averaged over 10 runs.

