# OpenReview forum: "Modelling Complex Tabular Datasets with a Mixture of Diverse Generative Models"
_TMLR — Accepted by TMLR_

### Review · Reviewer_c5t7 · 2026-03-05

**Summary Of Contributions:**

This paper proposes BIRD, a framework for modeling complex tabular datasets using a mixture of diverse generative models. The key idea is to assign different generators to different regions of the data space so that each generator specializes in a subset of the distribution. The paper also presents a theoretical analysis based on the robustness properties of the Maximum Mean Discrepancy (MMD), showing that under imperfect clustering, the learned mixture distribution can achieve an $O(n^{-1/2})$ error rate under certain assumptions. Empirical results on synthetic and real-world tabular datasets demonstrate improvements over several individual generators and mixture baselines.

Strengths:
1. The use of a bandit-inspired algorithm to select the most suitable generator for each cluster under a computational budget is an interesting design choice and distinguishes this work from prior mixture-of-generators approaches.
2. The paper provides a theoretical bound showing that if clusters are approximately recovered and generators are trained close to optimal within each cluster, the resulting mixture distribution can achieve a convergence rate comparable to the well-specified case.

Weaknesses:
1. The proposed framework fundamentally assumes that the data distribution can be decomposed into clusters that correspond to meaningful generative regions. However, many real-world tabular datasets may not exhibit well-separated cluster structures; instead, heterogeneity may appear as continuous density variations or conditional relationships. In such cases, it is unclear whether the clustering-based decomposition aligns with meaningful generative components. The paper does not provide analysis or experiments evaluating performance in weak-cluster or no-cluster regimes.
2. The theoretical results are derived for mixtures optimized under the MMD objective, assuming that clustering errors are bounded. However, the EM-like refinement step updates cluster memberships using an Energy score–based similarity rather than the MMD objective used in the theory. The paper does not establish that this update rule reduces clustering error or improves the MMD objective, leaving a gap between the theoretical guarantees and the actual optimization procedure.
3. The algorithm requires repeated computation of similarity scores between all data points and samples generated from each generator, which may scale poorly with dataset size and number of clusters. In addition, the reported computational budget appears to account only for generator training time while excluding the cost of computing membership probabilities. As a result, the true computational overhead of the method may be underestimated. Experiments are conducted only on relatively small tabular datasets, leaving scalability to larger datasets unclear. Also, the number of clusters appears to be determined implicitly by the initial clustering algorithm. However, the paper does not describe how the choice of cluster number is selected.

**Additional Comments:**

In Appendix F.3.1 (Credit Card Fraud Detection Dataset), the text refers to “Table ??” multiple times (e.g., “The experiments summarized in Table ??” and “The results presented in Table ??”). The table number appears to be missing and should be properly referenced.

**Audience:**

Yes

**Audience Explanation:**

This paper proposes one framework that allows different types of generative models (e.g., GANs, VAEs, diffusion models, random forests) to coexist within a mixture, enabling each model to specialize in different parts of the data space.

**Broader Impact Concerns:**

N.A.

**Claims And Evidence:**

Yes

**Claims Explanation:**

The paper provides both theoretical analysis and empirical evaluation to support its claims. This analysis provides useful insight into the robustness of mixture learning under imperfect clustering. The empirical results also demonstrate improvements over several baseline methods on multiple tabular datasets.

**Requested Changes:**

1. Provide additional experiments analyzing the sensitivity of the method to the number of clusters.

2. It would be interesting to run some experiments on datasets with weak or continuous structure to evaluate whether the clustering assumption is critical for performance.

3. Clarify the relationship between the Energy score–based update rule and the MMD-based theoretical analysis.

---

> ### Author Response · Authors · 2026-03-20
>
> **Weaknesses:**
>
>  1) The theoretical section assumes that the data has a meaningful clustering structure. However, our algorithm can be applied to any tabular dataset without specific assumptions. In our experiments, we utilize real-world data that might not always naturally partition into distinct clusters. By employing dimensionality reduction plots and calculating silhouette scores for various numbers of clusters in Section G of the appendix of the revised paper, we show experimentally that both the MiniBooNE\_PID and Wilt datasets exhibit weak or nonexistent clustering structures. Nevertheless, our BIRD algorithm performs well on these datasets, demonstrating that BIRD does not require a clear clustering structure to be effective. Although we cannot guarantee that the clustering-based decomposition directly corresponds to meaningful generative components, empirical evidence suggests a strong relevance.
>
>
>  In general, we anticipate improved performance in scenarios where data naturally separates into clusters. For instance, as shown in Appendix Section D, when we randomly shift data points between clusters after initial clustering, the performance of BIRD decreases. However, we think that many real-world datasets exhibit some implicit clustering structure.
>
>
> 2) We agree with the reviewer that our theoretical results does not apply exactly for the algorithm that we have proposed and we emphasize this further in the revised paper.
>
> In the theoretical results, we assume the clusters to be fixed and to be well enough approximated in order to be able to use robustness results of the MMD to prove some consistency bound. However, as mentioned in the paper, in practice it is a difficult to identify even approximate clusters, which is why we propose an algorithm that iteratively refines the clusters and generators simultaneously.
>
> To summarize, the theoretical results are provided in a simpler setting as the proposed algorithm and we acknowledge for this gap but we think that our theoretical results gives insights about the reason why we developed such an algorithm and we show the efficiency of the algorithm in practice. We added Remark 4.2 in the revised version of the paper to clarify this point.
>
> 3) We appreciate the reviewer's insight regarding the challenges of scaling the algorithm to larger datasets, especially in computing similarity scores. To address this, we have proposed strategies such as sub-sampling, as detailed in Section H.4 of the appendix in the revised paper. Our experiments demonstrate that the BIRD algorithm can be effectively applied to larger datasets through sub-sampling to compute similarity scores while maintaining strong performance. We added an experiment on the California Housing Prices dataset in Section H.4 as well.
>
> To illustrate the trade-off between computational efficiency and performance, we plan to perform sub-sampling at various ratios using the Wilt dataset, highlighting the time savings versus performance losses. We are currently conducting this experiment and can provide the results upon request. As noted in Appendix C, which details the algorithm's complexity, the computation of membership probabilities scales linearly with the number of clusters and dataset size. However, it is only quadratic with respect to the number of samples from generators.
>
> The number of clusters is automatically determined by the initial clustering method. We use the silhouette score, as detailed in Appendix H, to identify the optimal number of clusters.
>
> **Requested Changes**
>
> 1) We conduct experiments using simulated data, progressively increasing the number of clusters while maintaining the same value range. We then compare the performance of the BIRD algorithm for different number of clusters, in Section J of the appendix.  From these results, it appears that across various metrics, the performance of BIRD remains relatively stable as the number of clusters increases.
>
> 2) By employing dimensionality reduction plots and calculating silhouette scores for various numbers of clusters in Section G of the appendix of the revised paper, we show experimentally that both the MiniBooNE\_PID and Wilt datasets exhibit weak or nonexistent clustering structures. Nevertheless, our BIRD algorithm performs well on these datasets, demonstrating that BIRD does not require a clear clustering structure to be effective. We refer to the newly added section G of the appendix in the revised version of the paper for details.
>
> 3) As previously mentioned, the theoretical results differ from the proposed algorithm, see Remark 4.2 of the revised paper. Furthermore, it's important to note that the energy score helps determine each point's importance in training each generator, but it does not update the generator parameters. In the theoretical results, the MMD is used as a metric to evaluate the quality of the generators and we use its robustness to derive finite sample bounds.

---

> > ### Author Response · Authors · 2026-03-25
> >
> > We kindly inform the reviewer that we have added Section L, titled "Sensitivity to Random Sub-Sampling During Membership Probability Computation," to the appendix of the revised paper. This new section demonstrates the trade-off between computational efficiency and performance by employing random sub-sampling with various proportions using the Wilt dataset, highlighting the balance between time savings and performance losses.

---

### Review · Reviewer_GhFj · 2026-03-05

**Summary Of Contributions:**

This work proposes a method for generative modelling of tabular datasets with specific focus on the fact that they are multi-modal. There is assumed to be available a bag of available classes of generative models (e.g. normalizing flows, TabDDPM, adversarial random forests, etc.). Given a tabular dataset, the goal in this work is design a good model for the data by trying to simultaneously achieve all of the following: (a) clustering the data, (b) choosing the best class of generative model for each cluster, and (c) fitting the generative models to their cluster. This work's method, BIRD, achieves this in three steps (zero-indexed to match the paper's numbering):

0. (Clustering) Select an initial set of clusters (e.g., by K-means).
1. (Bandit-inspired algorithm) For each initial cluster, select a class of generative model by framing model selection as a stochastic multi-armed bandit problem where each round is an allotted amount of training time per model and the reward is model performance (inverse MMD). An algorithm from past work (R-SR) can be applied here.
2. (EM-inspired algorithm) Once a class of generator has been assigned to each cluster in step 2, they are then trained and clusters are adjusted using an EM-like algorithm modified from some past work on this problem, Banijamali et al. (2017).

The authors mainly focus on 2 datasets, and compare several popular baselines across a collection of metrics. Their method posts solid performance. The work's novelty is in building an end-to-end, highly flexible pipeline for fitting mixture models on tabular data, and to my understanding, the multi-armed bandit approach to model selection is a particularly innovative insight.

**Audience:**

Yes

**Audience Explanation:**

Yes; in particular, those interested in tabular generative models and mixtures of generative models. Assuming the issues below are addressed and the method still looks good, there is definitely some contribution here. As I mention above, I believe the main contribution is the multi-armed bandit approach to model selection.

**Broader Impact Concerns:**

No concerns

**Claims And Evidence:**

No

**Claims Explanation:**

What makes the method's efficacy not fully convincing in my opinion is in the baselining and ablations.

Here is my understanding of the baselining. I am most interested in the mixture baselines, since they are the most comparable to BIRD. In particular, these 3, which all begin with clustering as in BIRD but then perform the following
- **B1** Naive mixture - (1) a random, single universal class of generator is chosen for all clusters (2) generators are directly trained on their corresponding cluster
- **B2** Banijamali et al. (2017) - (1) a random, single universal class of generator is chosen for all clusters (2) they are trained using the EM-like algorithm of Banijamali et al. (2017)
- **B3** Random mixture - (1) a random, potentially different generator is chosen for each cluster (2) same EM-like approach as BIRD
-  In contrast, BIRD uses a bandit approach for (1) and its own EM-like approach for step (2) (similar to but different than Banijamali et al. (2017).

BIRD outperforms all of these on average by most metrics, but it's hard to disentangle why this happens for a few reasons:
1. There is no direct comparison between BIRD step (2) and Banijamali et al. step (2), which are purportedly very similar.
    - One approach would be to create a new ablation subbing BIRD step (2) for Banijamali et al. step (2) in **B2**.
    - Banijamali et al. step (2) is not described in detail in the work, but this work mentions that it is limited to homogeneous generators. Whether this is a fundamental limitation of the method, or just something that Banijamali et al. did not try, wasn't clear to me from the work. *If the latter is true and it is possible to straightforwardly modify Banijamali et al. to work for heterogeneous generators*, I would expect two additional experiments: a variant of **B3** using Banijamali et al for step (2), and a variant of BIRD that uses Banijamali et al (2017) as step (2) with the bandits approach for step (1).
2. Both **B1** (naive mixture) and **B2** (Banijamali et al.) start by initially sampling a random model class which will be used for all clusters. The variability in the quality of model classes (e.g. TabDDPM, CTGAN, etc.) makes their error bars absolutely gigantic! In both cases their error bars often overlap with BIRD or whatever else the top model is. This raises the possibility that just (1) choosing one of the best generators, e.g. ARF or TabDDPM, and using that for every cluster, and (2) training naively or training with Banijamali et al., would be a better approach than BIRD entirely.

**Requested Changes:**

**Required changes**
- (Baselining and ablations) As per my discussion above, we need to see more experiments to get a full picture of how well the model is performing and why. In particular, the following mixture experiments:
   1. (1) A random, single class of generator for all clusters + (2) BIRD step 2
   2. (If possible) (1) BIRD step 1 + (2) Banijamali et al. step 2
   3. (If possible) (1) A randomly chosen generator per cluster + (2) Banijamali et al. step 2
   4. To disentangle the performance of each generator, I would want to see a separate 10 run average + standard deviation for each generator as follows:
        1. (1) Use the same generator for each cluster + (2) train naively.
        2. (1) Use the same generator for each cluster + (2) train with Banijamali et al. step 2.
        3. (1) Use the same generator for each cluster + (2) train with BIRD step 2.
   5. It would also be very useful as a baseline to see how a hyperparameter-tuned SMOTE performs on these datasets as a baseline.

With these experiments in hand, it'll be much easier to understand how necessary the 2 steps of BIRD are to get good performance.

**Minors**
- I think the last inequality of the statement Corollary 3.5 and its proof in Appendix A needs to be flipped.
- Appendix F.3.1 has broken references to Table 10.
- Table 5 typo: "Benjamili’s mixture"
- App F.1: It'd be clearer if "Both ... randomly select one generator to assign to each initial cluster." were changed to "every initial cluster" or "all initial clusters" to clarify that the same generator is used.

Question: If I'm understanding correctly, Assumption 3.4 seems trivially true for any $\chi > 0$ by the definition of infimum. So it is only non-trivial in the $\chi=0$ case. Is it supposed to serve as more of a definition of $\chi$? Would you mind clarifying if I'm understanding correctly and why/when it's important?

---

> ### Author Response · Authors · 2026-03-20
>
> - *"There is no direct comparison between BIRD step (2) and Banijamali et al. step (2), which are purportedly very similar"*:
> The second step of BIRD introduces novelty compared to Banijamali's step (2), such as the selection of similarity measures and incorporating point importance through the choice of mini-batches for training, we added Remark 4.1 in the revised paper to clarify this point. We agree with the reviewer's observation regarding the lack of a direct empirical comparison between BIRD step (2) and Banijamali et al. step (2). We believe incorporating such a comparison, as proposed by the reviewer, would be highly beneficial. Therefore, we have added additional baselines to perform this comparison in Section H.5 of the revised paper.
>
> - *"Banijamali et al. step (2) is not described in detail in the work, but this work mentions that it is limited to homogeneous generators"*: The work of Banijamali is limited to a single class of generators, in their case Neural networks with a given architecture. At each iteration of their Steop (2), the parameters of the network are learned by using a gradient step with the MMD as the loss function. This method could potentially be extended to various types of generators, but provided that they allow for gradient updates using an MMD loss as part of their training procedure. Our BIRD algorithm is versatile and can accommodate any generators with any training method. If necessary, we can include a remark in the paper to clarify this point.
>
> - *"This raises the possibility that just (1) choosing one of the best generators, e.g. ARF or TabDDPM, and using that for every cluster, and (2) training naively or training with Banijamali et al., would be a better approach than BIRD entirely."*: We acknowledge the reviewer's concern regarding the variability in model class quality when initially sampling a random model class for all clusters. However, selecting the single best generator for use across all clusters assumes prior knowledge of which generator performs optimally for a specific problem—knowledge that is often unavailable in practice. Therefore, comparing this approach with our BIRD algorithm, which doesn’t rely on prior assumptions, may not be entirely fair. Nonetheless, we implemented the baselines suggested by the reviewer on the Wilt dataset in section H.5 of the appendix. This involved: (1) selecting a top-performing generator, such as ARF or TabDDPM, to be used universally across clusters, and (2) conducting training either naively or with the methods proposed by Banijamali et al., or employing the second step of BIRD. We see that BIRD remains the best-performing method, indicating that none of the competitors using a single generator for all clusters outperforms it, which shows the importance of mixing generative models. In other words, assigning different types of generators to different clusters can help improve performance.
>
>
> **Requested changes**
>
> We refer to the section H.5 of the appendix of the revised for the experiments $1$ to $4$ and the analysis of the results. Overall, these experiments suggest the superiority of BIRD’s step (2) over Banijamali's step (2) and naive training.
>
> 5. SMOTE experiment:
>
> The SMOTE algorithm generates samples from minority classes and assumes the presence of a variable in the dataset indicating the class of each observation. In contrast, our BIRD algorithm can generate data from any dataset without requiring class labels. To compare our algorithm with SMOTE, we used the WILT dataset, which contains class information. We generated data using the SMOTE implementation from the *imbalanced-learn* package and conducted hyperparameter optimization with the *optuna* package in Python. We generated 100 simulated datasets and present the results in the table below.
>
> | Generator Type | MMD (1e-3) ↓ | JSD (1e-3) ↓ | IKL (1e-1) ↑ | P-PR ↑ | P-RE ↑ |
> |----------------|--------------|--------------|--------------|--------|--------|
> | **BIRD**       | 0.717 ± 0.254| 3.152 ± 0.472| 9.54 ± 0.149 | 0.784 ± 0.0077 | 0.8936 ± 0.005 |
> | **SMOTE**      | 1.358 ± 0.0756 | 8.572 ± 0.284 | 9.770 ± 0.098 | 0.957 ± 0.0025 | 0.9372 ± 0.0005 |
>
> The similarity measures in the table indicate that synthetic datasets generated using the SMOTE algorithm are more closely aligned with real datasets than those generated by BIRD for several metrics, including IKL, P-PR, and P-RE. However, because SMOTE relies on Nearest Neighbors, it poses privacy concerns by potentially exposing private information from the real dataset. This issue is highlighted in the recent paper by Ganev et al., ICLR 2026, "SMOTE and Mirrors: Exposing Privacy Leakage from Synthetic Minority Oversampling". SMOTE is likely to reveal private information from the real dataset, which is a crucial concern in many fields such as medicine.
>
> **Question**: We have modified Assumption 3.4 in the revised version of the paper to take this question into account and implement the minor comments.

---

> > ### Comment · Reviewer_GhFj · 2026-03-27
> >
> > Thank you to the authors for running the additional experiments I've requested. I'm glad to see BIRD still performs will against these baselines. This satisfies my requested changes.

---

### Review · Reviewer_yAM1 · 2026-03-07

**Summary Of Contributions:**

The paper proposes **BIRD**, a framework for modeling complex tabular datasets using a **mixture of heterogeneous generative models**. The method combines (1) a **bandit-based procedure** that assigns generators to clusters under a computational budget and (2) an **EM-style refinement** step that iteratively updates cluster memberships based on similarity between real data points and samples generated by each model. Generators are then retrained using minibatches sampled according to these membership probabilities. The approach is designed for generators with intractable likelihoods, relying only on sampling. Experiments on tabular datasets suggest improvements over individual generators and simple mixture baselines.

### Strengths
- Addresses a **relevant problem** in tabular generative modeling, where single generators often struggle with multi-modal data distributions.

- The **bandit-based generator assignment** is an interesting idea that helps allocate computational resources efficiently and select suitable generators for each cluster.


### Weaknesses

- **Experimental evaluation is relatively small-scale**, using small tabular datasets; evaluation on larger and more diverse datasets would strengthen the empirical evidence.
- Potential **scalability concerns**, since computing membership probabilities requires similarity evaluations between real and generated samples.
- The algorithm **assumes that the data admits a meaningful clustering structure**, which may not hold for many real-world tabular datasets where clusters are weak, overlapping, or not well-defined.
- **Limited methodological novelty**: the EM-style refinement step is closely related to prior work on mixtures of generative models, with the main novelty primarily coming from the bandit-based initialization.

**Audience:**

Yes

**Audience Explanation:**

The problem of generating high-quality synthetic tabular data is relevant to researchers working on generative models and tabular data, and the idea of combining heterogeneous generators in a mixture framework may therefore interest part of the TMLR audience. In particular, the bandit-based generator assignment is a potentially interesting component. However, the overall novelty of the method appears somewhat limited, especially since the refinement procedure is closely related to prior mixture-of-generators approaches, which may reduce the broader impact of the work.

**Broader Impact Concerns:**

No significant ethical concerns are apparent beyond those commonly associated with synthetic data generation. The proposed method aims to generate synthetic tabular data, which can potentially be used for privacy-preserving data sharing and benchmarking. However, as with other generative models, there is a general risk that synthetic data could unintentionally reproduce sensitive patterns from the original dataset or be misused if deployed without appropriate safeguards.

These considerations are standard for synthetic data generation methods and do not appear to introduce new ethical risks specific to this work. Overall, no additional broader impact discussion seems strictly necessary beyond acknowledging the general considerations related to synthetic data use and privacy.

**Claims And Evidence:**

Yes

**Claims Explanation:**

The main claims of the paper are **partially** supported by the evidence provided. The experimental results show improvements over several baselines on the evaluated datasets, and the paper includes both empirical and theoretical arguments supporting the approach. However, the experiments are conducted on relatively **small tabular datasets**, which limits the strength of the empirical validation. In addition, while the paper discusses computational complexity and clustering robustness in the appendices, further empirical evaluation (e.g., **scalability analysis and experiments on larger benchmark datasets**) would strengthen the support for the broader claims about the method’s effectiveness and practicality.

**Requested Changes:**

**Critical for acceptance**

- **Strengthen the empirical evaluation.**
  The current experiments are conducted on relatively small tabular datasets. To better demonstrate the practical relevance of the approach, it would be important to evaluate the method on larger and more widely used tabular benchmarks (e.g., datasets commonly used in recent tabular generative modeling studies). Such experiments would help assess whether the proposed framework scales to more realistic scenarios and whether the performance gains remain consistent as dataset size and dimensionality increase. Additionally, including more diverse datasets (in terms of size, feature types, and distribution complexity) would strengthen the empirical support for the paper’s claims.

- **Provide additional scalability analysis.**
  The proposed method requires computing membership probabilities by measuring similarity between real data points and generated samples from each generator. This procedure may become computationally expensive as the dataset size, number of clusters, or number of generated samples increases. While the appendix discusses the computational complexity, it would be useful to complement this analysis with empirical runtime measurements. For example, the authors could report training time as a function of dataset size or number of clusters, and evaluate whether the proposed subsampling strategies effectively mitigate the computational overhead in practice.

- **Further analyze sensitivity to clustering assumptions.**
  The algorithm relies on the assumption that the dataset can be meaningfully partitioned into clusters that correspond to different generators. Although the appendix includes an experiment where cluster assignments are artificially degraded, additional analysis would help clarify the robustness of the approach. For example, experiments comparing different clustering algorithms (e.g., k-means, hierarchical clustering, density-based clustering) or evaluating datasets with weaker or overlapping cluster structures would provide more insight into how sensitive the method is to the initial clustering step.

**Suggestions that would strengthen the work**

- **Clarify the methodological novelty.**
  The EM-style refinement procedure appears conceptually related to earlier mixture-of-generators approaches. While the bandit-based generator assignment is an interesting addition, the paper would benefit from a clearer discussion of how the proposed refinement mechanism differs from existing methods. In particular, an ablation study isolating the contribution of the bandit initialization versus the iterative refinement stage would help clarify which components of the framework are primarily responsible for the observed performance improvements.

- **Expand the discussion of hyperparameter choices.**
  The framework introduces several design choices, including the number of clusters, the pool of candidate generators, the similarity metric used for computing membership probabilities, and the allocation of computational budget between algorithm stages. Additional discussion or experiments analyzing the sensitivity of the method to these parameters would improve the clarity and usability of the approach. Providing practical guidelines or heuristics for selecting these parameters would also help practitioners apply the method in real-world settings.

---

> ### Author Response · Authors · 2026-03-20
>
> **Weaknesses**
> - "*Experimental evaluation is relatively small-scale*": While we acknowledge the reviewer's observation that our experiments primarily utilize small-scale datasets due to the computational cost of calculating similarity measures, we address this concern in Appendix H.4. There, we demonstrate an experiment on a larger dataset ($20\,000$ points), showing that our method can effectively scale to larger problems by randomly sub-sampling points during similarity calculations. The BIRD algorithm maintains strong performance with random sub-sampling when compared to baseline methods. Additionally, we conducted experiments on datasets such as MiniBooNE, Wilt, which are also benchmarked in other tabular generative method studies (Kotelnikov et al., 2024, "TabDDPM: Modelling Tabular Data with Diffusion Models").
>
>     Additionally, we conducted an experiment on the California Housing dataset, which includes 20,640 observations, as used in Kotelnikov's paper. The results are presented in the Section H.4 of the paper.
>
> -  "*Potential scalability concerns*":
>     We agree with the reviewer that computing membership probabilities is resource-intensive, as it requires similarity evaluations between real and generated samples. To address this, we explored several alternatives. In Section H.4 of the appendix, we propose using random sub-sampling for computing similarity measures, which has shown empirical effectiveness with BIRD still outperforming its competitors. However, there may be more optimal sub-sampling methods, which we are eager to discuss further in the paper.
>
>     Additionally, we investigated modifying the similarity measure itself. We conducted an experiment on the Wilt dataset, implementing the BIRD algorithm with alternative distances, such as Gaussian and Euclidean distances, in place of the energy score. While these alternatives are quicker to compute and significantly reduce the computational burden, they lead to a decline in algorithm performance compared to the energy score. See section K of the appendix.
> Finally, we have provided a detailed analysis of the complexity of the algorithm in Section C of the appendix.
>
> - "*The algorithm assumes that the data admits a meaningful clustering structure*": The theoretical section assumes that the data has a meaningful clustering structure. However, our algorithm can be applied to any tabular dataset without specific assumptions. We operate under the implicit assumption that it is easier to divide the data space into regions and train separate generators for each, rather than directly learning a generator for the entire dataset. We demonstrate empirically that this approach often simplifies the training process for generators.  We added Remark 4.2 in the revised paper to emphasize this.
>
>     In Section G of the revised paper, we demonstrate through dimension reduction plots and Silhouette scores that the real datasets used lack a clear clustering structure. Specifically, the Wilt and MiniBooNE\_PID datasets exhibit weak or nonexistent clustering. Despite this, our BIRD algorithm performs well, highlighting its effectiveness even in the absence of a distinct clustering structure.
>
> - "*Limited methodological novelty:  the EM-style refinement step is closely related to prior work on mixtures of generative models*"
>
>     Our work presents a comprehensive end-to-end approach for training a mixture of generators with a fixed budget, directly from data with several novelties. The EM refinement step is similar to Banijamali's second step. However, BIRD's second step introduces novel elements, such as selecting similarity measures and incorporating point importance through the use of mini-batches for training. We have added Remark 4.1 in the revised paper to clarify this distinction.
>
> **Requested changes**
> -  We have conducted an additional time complexity analysis of our random sub-sampling approach, as suggested by the reviewer. We examine how the computational time change with variations in dataset size and the number of initial clusters. See section C.
>
> -  In Section G of the appendix, we demonstrate that the Wilt and MiniBooNE\_PID datasets have weak to nonexistent clustering structures. In our other experiments, BIRD still performs well on them.
>
> -  We have previously highlighted the methodological innovations of our paper and added Remark 4.1 in the revised document, which summarizes the novel aspects of our Step 2 compared to that proposed by Banijamali. To empirically evaluate the significance of each step in BIRD, we provide an additional baseline using the Wilt dataset, see Section H.5.
>
> - Sensitiviy to budget allocation (Section E), initial clustering (Section D), generator pools (Section H.6), and the sensitivity to similarity metrics (section K). The number of clusters is selected automatically using the Silhouette Score.
>
> -  We acknowledge the reviewer's remark on the general considerations related to synthetic data use and privacy.

---

> > ### Comment · Reviewer_yAM1 · 2026-04-07
> >
> > Thank you to the authors for the additional experiments and clarifications addressing my concerns. The revised version improves the paper in several aspects. In particular, the analysis of clustering assumptions is now clearer and supported by empirical evidence. The issues related to scalability and experimental validation have been partially addressed through experiments on larger datasets and the introduction of sub-sampling strategies. However, the approach still relies on similarity computations between real and generated samples, and the sub-sampling strategy introduces an approximation whose impact may vary across datasets and parameter choices. As a result, some uncertainty remains regarding how the method behaves as the dataset size and model complexity increase.

---

### Author Response · Authors · 2026-05-12

As the review period has concluded, we wanted to kindly ask if there are any updates regarding the decision.
We understand that the process can be time-consuming, and we really appreciate your efforts.

If there is any additional information or clarification needed from our side, please feel free to let us know.

Thank you very much!

---

### Decision · Action_Editor_17aF · 2026-05-16

**Recommendation:** Accept with minor revision

**Additional Comments:**

For the minor revision, I ask the authors to make the following final changes. The paper should make it sufficiently clear that the theoretical guarantees apply to the simplified fixed-cluster MMD setting analyzed in Section 3, and should not be interpreted as convergence or correctness guarantees for the full BIRD algorithm, where clusters and generators are iteratively updated together. Next, the discussion of scalability should be stated carefully, since the computation of membership probabilities remains a practical computational bottleneck even when subsampling strategies are used. Finally, the paper should briefly discuss the SMOTE comparison raised during the rebuttal phase, clarifying that BIRD is designed for general synthetic tabular data generation without requiring class labels, whereas SMOTE is a label-aware oversampling method that can perform strongly when labels are available.

**Audience:**

Yes

**Audience Explanation:**

Synthetic tabular data generation is a practically relevant problem with applications in privacy-preserving data sharing and data augmentation. Researchers working on generative modeling, mixture models, and tabular data will find the paper's contributions relevant, especially the application of a bandit-based algorithm for generator assignment under a fixed computational budget, and the theoretical analysis connecting MMD robustness to mixture learning under imperfect clustering.

**Claims And Evidence:**

Yes

**Claims Explanation:**

The paper’s empirical claims are largely supported by evidence across multiple datasets and metrics. The main claim that BIRD improves over individual generators and competing mixture methods is supported by experiments on several datasets, with results averaged over multiple independent runs and reported with standard deviations. In response to reviewer concerns, the authors added ablations to better disentangle the contributions of the bandit-based generator assignment step and the EM-type refinement step, including comparisons against variants with random generator assignment, single-generator choices, and alternative refinement/training procedures. Overall, these additional results provide convincing evidence that both components contribute to the method’s performance and that the empirical gains are not merely due to the presence of a particularly strong generator in the candidate pool.

The theoretical claim is more limited but reasonably supported within its stated scope. The paper shows that, under Assumptions 3.2–3.4 and a fixed approximately correct clustering, learning a mixture from imperfectly identified clusters can achieve an $O(n^{-1/2})$-type error rate. The proof in Appendix A relies on existing MMD generalization and robustness results, together with triangle-inequality arguments for kernel mean embeddings. The authors acknowledge in Remark 4.2 that this result does not directly establish a convergence guarantee for Algorithm 1, where clusters and generators are iteratively co-updated. This point should remain clear: the theory should be viewed as supporting intuition for the proposed framework rather than as a full correctness proof for BIRD. With this scope made explicit, I find the theoretical evidence acceptable.

The minority-class modeling results, including Tables 1 and 14, are among the strong empirical findings, as they suggest that BIRD can better capture underrepresented class distributions than several individual generators. The scalability claims are partially supported by the larger-dataset experiments in Appendix H and the subsampling sensitivity analysis in Appendix L, although scalability remains a practical limitation because membership-probability computation can still be costly. One caveat is that the SMOTE comparison reported in the rebuttal shows that, when class labels are available, SMOTE can perform better than BIRD on several metrics. This should be discussed more clearly in the final version to clarify that BIRD is mainly intended for general synthetic tabular generation without requiring class labels, rather than as a replacement for label-aware oversampling methods. Given these clarifications, I find the claims to be supported by sufficiently convincing and clear evidence for acceptance.

---

> ### Author Response · Authors · 2026-06-22
>
> Dear Action Editor,
>
>
> We would like to thank you and the reviewers for your valuable feedback. We have uploaded the camera-ready version of our manuscript, incorporating the requested minor revisions.
> In response to the comment asking us to “make it sufficiently clear that the theoretical guarantees apply to the simplified fixed-cluster MMD setting analyzed in Section 3,” we revised a sentence in the abstract to clarify that our theoretical results apply only to this simplified setting. We also added the following clarification in the Discussion:
>
>
> “However, it is important to clarify that our theoretical results are formulated for simplified settings and do not serve as guarantees of convergence or correctness for the complete BIRD algorithm, where clusters and generators are iteratively updated together.”
>
> In addition, Remark 4.2 further emphasizes this point.
>
>
> To address the comment regarding SMOTE, we added a short paragraph at the end of the Discussion and refer to a newly created section in the Appendix.
>
>
> Finally, regarding scalability, we added a paragraph in the Discussion noting that, although our subsampling results presented in the Appendix are promising, developing a systematic approach to address this challenge remains an open research question.
>
>
> We hope that these revisions satisfactorily address the comments. Please do not hesitate to contact us if any further changes are needed. Finally, we corrected small typos in the paper and developed each step of the Proof in Appendix A.

---

> > ### Comment · Action_Editor_17aF · 2026-06-28
> >
> > Dear Authors,
> >
> > Thank you for incorporating the requested minor revisions, which address the remaining reviewers’ comments. Since this submission is marked as a “Regular submission (no more than 12 pages of main content),” I ask that you upload a new version of the accepted paper in which the Discussion section ends by Page 12.
> >
> > AE

---

> > > ### Author Response · Authors · 2026-06-28
> > >
> > > Thank you, we have updated the camera-ready paper such that the discussion ends on page 12.